# Expansion and conversion of human pancreatic ductal cells into insulin-secreting endocrine cells

Jonghyeob Lee[1], Takuya Sugiyama[1†], Yinghua Liu[1†], Jing Wang[1], Xueying Gu[1], Ji Lei[2], James F Markmann[2], Satsuki Miyazaki[3], Jun-ichi Miyazaki[3], Gregory L Szot[4], Rita Bottino[5], Seung K Kim[1,6*]

[1]Department of Developmental Biology, Stanford University School of Medicine, Stanford, United States; [2]Department of Surgery, Massachusetts General Hospital, Harvard Medical School, Boston, United States; [3]Division of Stem Cell Regulation Research, Osaka University Graduate School of Medicine, Osaka, Japan; [4]UCSF Transplantation Surgery, University of California, San Francisco, San Francisco, United States; [5]Department of Pediatrics, Division of Immunogenetics, Children's Hospital of Pittsburgh, University of Pittsburgh School of Medicine, Pittsburgh, United States; [6]Department of Medicine, Oncology Division, Howard Hughes Medical Institute, Stanford University School of Medicine, Stanford, United States

*For correspondence:
seungkim@stanford.edu

†These authors contributed equally to this work

**Competing interests:** The authors declare that no competing interests exist.

**Reviewing editor**: Janet Rossant, University of Toronto, Canada

**Abstract** Pancreatic islet β-cell insufficiency underlies pathogenesis of diabetes mellitus; thus, functional β-cell replacement from renewable sources is the focus of intensive worldwide effort. However, in vitro production of progeny that secrete insulin in response to physiological cues from primary human cells has proven elusive. Here we describe fractionation, expansion and conversion of primary adult human pancreatic ductal cells into progeny resembling native β-cells. FACS-sorted adult human ductal cells clonally expanded as spheres in culture, while retaining ductal characteristics. Expression of the cardinal islet developmental regulators Neurog3, MafA, Pdx1 and Pax6 converted exocrine duct cells into endocrine progeny with hallmark β-cell properties, including the ability to synthesize, process and store insulin, and secrete it in response to glucose or other depolarizing stimuli. These studies provide evidence that genetic reprogramming of expandable human pancreatic cells with defined factors may serve as a general strategy for islet replacement in diabetes.

## Introduction

The pancreas is a vital organ with exocrine and endocrine cell functions, and a root of lethal human diseases including diabetes mellitus, pancreatitis, and pancreatic ductal adenocarcinoma. Exocrine acinar cells produce digestive zymogens that are delivered to the intestines by a branching network of exocrine ductal cells that secrete bicarbonate and other products. Pancreatic endocrine functions derive from clusters of epithelial cells (islets of Langerhans) called α-, β-, δ-, and PP-cells that respectively synthesize, store, and secrete the hormones Glucagon, Insulin, Somatostatin, and Pancreatic polypeptide (*Benitez et al., 2012*). Insulin production by islet β-cells is highly regulated: key features of mature β-cells include preproinsulin (*INS*) transcription, proinsulin processing by endo- and exo-peptidases and storage of the proinsulin cleavage products insulin and C-peptide in dense core vesicles. Likewise, cardinal β-cell functions regulate insulin release in response to glucose and other secretagogues, including glucose sensing and metabolism through the enzyme glucokinase, and use of ATP-dependent potassium channels ($K_{ATP}$) and voltage-gated calcium channels to induce insulin

**eLife digest** Diabetes mellitus is a disease that can lead to dangerously high blood sugar levels, causing numerous complications such as heart disease, glaucoma, skin disorders, kidney disease, and nerve damage. In healthy individuals, beta cells in the pancreas produce a hormone called insulin, which stimulates cells in the liver, muscles and fat to take up glucose from the blood. However, this process is disrupted in people with diabetes, who either have too few pancreatic beta cells (type 1 diabetes) or do not respond appropriately to insulin (type 2 diabetes).

All patients with type 1 diabetes, and some with type 2, must inject themselves regularly with insulin, but this does not always fully control the disease. Some type 1 patients have been successfully treated with beta cells transplanted from deceased donors, but there are not enough donor organs available for this to become routine. Thus, intensive efforts worldwide are focused on generating insulin-producing cells in the lab from human stem cells. However, the cells produced in this way can give rise to tumors.

Now, Lee et al. have shown that duct cells, which make up about 30% of the human pancreas, can be converted into cells capable of producing and secreting insulin. Ductal cells obtained from donor pancreases were first separated from the remaining tissue and grown in cell culture. Viruses were then used to introduce genes that reprogrammed the ductal cells so that they acquired the ability to make, process and store insulin, and to release it in response to glucose—hallmark features of functional beta cells.

As well as providing a potential source of cells for use in transplant or cell conversion therapies for diabetes, the ability to grow and maintain human pancreatic ductal cells in culture may make it easier to study other diseases that affect the pancreas, including pancreatitis, cystic fibrosis, and adenocarcinoma.

exocytosis (reviewed in *Suckale and Solimena, 2010*). Deficiency or malfunctioning of β-cells produces impaired glucose regulation and diabetes mellitus, a disease with autoimmune (type 1, T1DM) and pandemic forms (type 2; *Ashcroft and Rorsman, 2012*). Thus, replacement or regeneration of functional human β-cells is an intensely-sought goal.

Human islet transplantation can be used to replace β-cell function in T1DM (reviewed in *Vardanyan et al., 2010*), but a shortage of donors currently precludes broad use of human pancreatic islets for β-cell replacement. Because of their expandability and multipotency, human embryonic stem cells (hESCs) and induced pluripotent stem cells (iPSCs) have been explored as sources of replacement insulin-producing cells (reviewed in *Hebrok, 2012*). However, directing the differentiation of these developmentally 'primitive' cells through multiple sequential fates into β-cell-like progeny that synthesize, process, store, and secrete insulin while lacking tumorigenic potential has challenged investigators worldwide (*Fujikawa et al., 2005*; *McKnight et al., 2010*; *Cheng et al., 2012*). Moreover, different hESC and iPSC cell lines exhibit significant variability during development into insulin-producing cells (*Nostro and Keller, 2012*). Recent work demonstrated that differentiated cell types in adult organs, including the mouse pancreas, can be experimentally 'reprogrammed' into progeny resembling islet cells, suggesting a new strategy for β-cell replacement (*Vierbuchen and Wernig, 2011*). For example, adult mouse pancreatic acinar cells can be converted into insulin-producing cells in vitro and in vivo (*Minami et al., 2005*; *Zhou et al., 2008*). However, little progress has been made in reprogramming primary human epithelial cells into different cell types, including conversion of pancreatic non-β-cells toward a human β-cell fate (*Vierbuchen and Wernig, 2011*). Thus, systems permitting expansion and genetic modulation of human pancreatic cells could powerfully influence studies of β-cell biology and replacement.

Pancreatic ducts constitute 30–40% of human pancreas and have been proposed as a potential source of replacement β-cells (*Bouwens and Pipeleers, 1998*; *Bonner-Weir et al., 2004*). During pancreas development, fetal endocrine cells derive from primitive ductal epithelium (reviewed by *Pan and Wright, 2011*; *Pictet and Rutter, 1972*). In addition, some studies have suggested that in adult mice, β-cells may be produced from pancreatic ductal epithelium (*Inada et al., 2008*; *Xu et al., 2008*; *Rovira et al., 2010*). However, recent lineage tracing evidences have not supported this view (*Solar et al., 2009*; *Furuyama et al., 2011*; *Kopp et al., 2011*). In humans, prior studies have suggested that adult human primary ductal cells in heterogeneous cell mixtures may harbor the potential

to generate endocrine-like progeny (*Bonner-Weir et al., 2000*; *Heremans et al., 2002*; *Swales et al., 2012*), but interpretation in these studies was limited by the probability of islet cell contamination. Therefore, the potential for conversion of pancreatic ductal cells toward an endocrine fate remains unclear. Moreover, prior studies have revealed only limited proliferative capacity of primary human pancreatic ductal cells in culture (*Rescan et al., 2005*). Thus, despite their relative abundance, multiple practical issues have prevented development of human pancreatic ductal cells as a source of replacement β-cells.

Here we report that normal human adult pancreatic duct cells can be sorted, clonally expanded, and genetically converted into endocrine cells. Human insulin-producing cells (IPCs) produced from sorted duct cells exhibited hallmark features of functional neonatal β-cells including high-level preproinsulin (*INS*) expression, proinsulin processing and dense-core granule formation. Moreover, secretion of insulin and insulin C-peptide from IPCs is stimulated by glucose and $K_{ATP}$ channel stimulants in a calcium-dependent manner. Together these studies reveal a new system for investigating human pancreatic duct cell biology, genetics, and β-cell regeneration.

## Results

### Purification and expansion of primary CD133[+] human pancreatic ductal cells

To identify human pancreatic epithelial cells that can be grown and maintained in culture, we systematically screened cell isolation methods and culture conditions with dispersed adult human pancreatic cells obtained from cadaveric donors without known pancreatic cancer, diabetes mellitus, or other pancreatic diseases (*Table 1*). With primary cells plated at low density, we observed formation of multicellular epithelial spheres, when cultured in Matrigel with a serum-free culture medium without feeder cells ('Materials and methods', *Figure 1—figure supplement 1A*). The multicellular sphere formation suggested primary cell expansion, so based on this assay we fractionated cells by fluorescence-activated cell sorting (FACS) to isolate and characterize sphere-forming pancreatic cells. A survey of cell surface markers used for fetal mouse pancreatic cell isolation (*Sugiyama et al., 2007*) revealed that antibodies recognizing CD133 enriched sphere-forming cells by four fold, whereas sphere-forming cells were depleted in the CD133[neg] fraction (*Figure 1A,B*). Immunohistochemical analysis of the human adult pancreas revealed CD133 expression at the apical portion of duct epithelial cells that co-expressed keratin 19 (KRT19), whereas CD133 was undetectable in islet endocrine cells or acinar cells (*Figure 1C*, *Figure 1—figure supplement 1B*), consistent with prior reports (*Lardon et al., 2008*). We have achieved sphere formation from over 35 consecutive adult donors (*Table 1*); thus, the sphere formation of primary adult human pancreatic CD133[+] cells was highly reproducible.

To assess the properties of FACS-purified adult pancreatic CD133[+] cells, we performed quantitative reverse transcription PCR (qRT-PCR). This revealed that CD133[+] cells expressed high levels of mRNA encoding ductal markers (*KRT19* and *CAR2*), while mRNAs expressed in acinar (*CPA1* and *CEL*) and endocrine (*CHGA, INS,* and *GCG*) cells were exclusively enriched in the CD133[neg] fraction (*Figure 1D*, *Figure 1—figure supplement 1C*). Immunostaining confirmed that >98% of sorted CD133[+] cells produced KRT19, whereas CD133[+] cells produced no detectable islet hormone (*Figure 1E,F*, *Figure 1—figure supplement 1D*). Thus, FACS efficiently eliminated islet endocrine and acinar cells, and enriched for a population of primary adult pancreatic duct cells that expanded as epithelial spheres in feeder- and serum-free culture.

### Maintenance of ductal phenotypes by self-renewing human pancreatic CD133[+] cells

After commencing in vitro cultures, the epithelial spheres from CD133[+] ductal cells attained diameters ranging from 40 to 520 μm in 2 weeks (*Figure 1A*, *Figure 1—figure supplement 1A* and *Figure 2—figure supplement 1A*). Spheres 350–500 μm in diameter were composed of 1470 ± 310 cells (n = 5); thus, based on evidence of clonal expansion (see below), we calculated that spheres resulted from a minimum of 10 cell divisions in 2 weeks. Sphere epithelium maintained KRT19 protein expression and a polarized monolayer as indicated by apical localization of CD133 (*Figure 2A*, *Figure 2—figure supplement 1A,D*). Neither acinar (CPA1) nor islet endocrine (CHGA and insulin C-peptide) markers were detectable (*Figure 3C* and data not shown), suggesting epithelial cells in cultured spheres maintained ductal characteristics.

**Table 1.** Phenotypes of pancreas donors

| Anonymous ID | Age (year) | Gender | Body mass index |
|---|---|---|---|
| 1 | 31 | Male | 28.1 |
| 2 | 52 | Male | 31.6 |
| 3 | 52 | Male | Not provided |
| 6 | 16 | Female | 20.4 |
| 9 | 34 | Male | 35.4 |
| 10 | 50 | Female | 23 |
| 11 | 32 | Female | 36.2 |
| 12 | 35 | Male | 45.7 |
| 13 | 23 | Female | 26.6 |
| 14 | 51 | Female | 23.3 |
| 15 | 48 | Male | 36.7 |
| 16 | 25 | Male | 21.8 |
| 17 | 63 | Female | 30.9 |
| 18 | 44 | Male | 24.7 |
| 19 | 39 | Male | 27.36 |
| 20 | 44 | Male | 23.5 |
| 21 | 50 | Female | 31 |
| 22 | 40 | Female | 26 |
| 23 | 53 | Male | 31 |
| 24 | 19 | Female | 20.83 |
| 25 | 34 | Male | 22.8 |
| 26 | 55 | Male | 37.7 |
| 27 | 17 | Female | 31.1 |
| 28 | 33 | Male | 18.8 |
| 29 | 48 | Male | 36.6 |
| 30 | 40 | Female | 28.4 |
| 31 | 43 | Female | 35.3 |
| 32 | 47 | Female | 21 |
| 33 | 48 | Female | 23.3 |
| 37 | 28 | Male | 24.2 |
| 40 | 34 | Male | 32.8 |
| 41 | 22 | Male | 19.6 |
| 42 | 53 | Female | 22.4 |
| 44 | 16 | Male | 33.9 |
| 45 | 54 | Male | 29.6 |
| 46 | 18 | Male | 21.8 |
| 48 | 24 | Male | 25.5 |

To assess whether sphere growth was achieved by cell proliferation or by other mechanisms like cell migration and aggregation, we analyzed spheres by immunostaining and time-lapse imaging. Immunohistochemistry revealed the proliferation marker Ki-67 in more than 25% of cells comprising 2-week-old spheres (*Figure 2A*, *Figure 2—figure supplement 1B*; labeling index 26.5 ± 5.1%), data further supported by detection of a second proliferation marker, phospho-histone H3 (*Figure 2A*). Time-lapse imaging revealed that spheres arose from single cells (*Figure 2B*), providing strong evidence that sphere formation resulted from CD133$^+$ ductal cell proliferation, rather than through cell migration and aggregation. Enzymatic dispersion of 2-week-old G1 spheres and subsequent culture

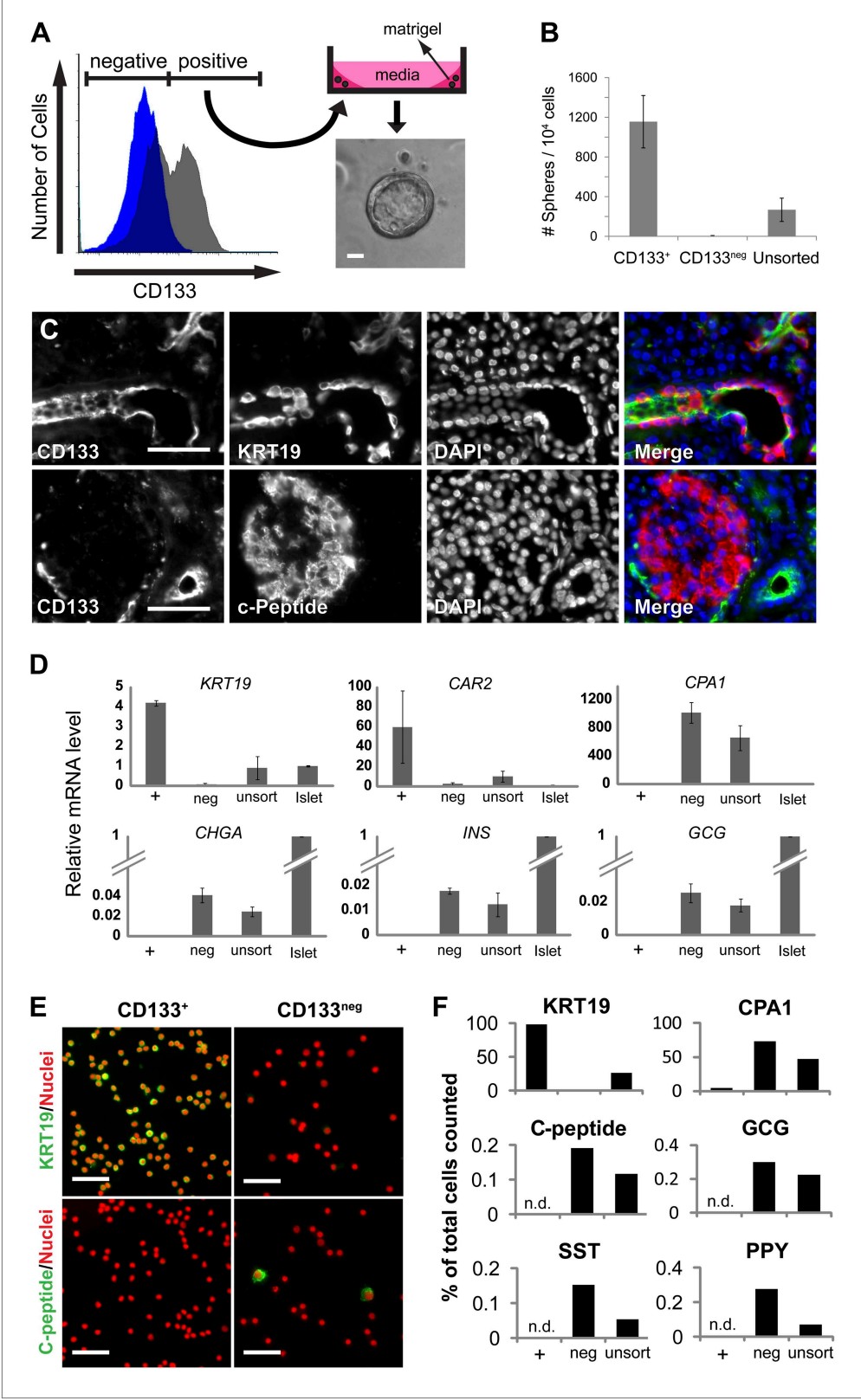

**Figure 1**. The ductal cell surface marker CD133 enriches sphere-forming cells from dissociated human adult pancreas. (**A**) Left panel, FACS plot of the dissociated human adult pancreas stained with (gray) or without (blue) antibodies specific for CD133. Right panel, A schematic of the sphere culture system and a representative sphere

*Figure 1. Continued on next page*

*Figure 1. Continued*

after culture. (**B**) Quantification of spheres generated from CD133$^+$, CD133$^{neg}$, and unsorted cells. Data are presented as mean ± SEM (n = 4). (**C**) Immunostaining of CD133 (green) with a ductal marker KRT19 (red) and C-peptide (red) in adult human pancreas. (**D**) The gene expression profiles of FACS-sorted human adult pancreatic cells and isolated islets (islet values normalized to 1). Data are presented as mean ± SEM (n = 3). (**E**) Representative immunostaining pictures of sorted cells with KRT19 (green) or C-peptide (green). (**F**) Quantification of cell immunostaining after FACS. ≥7200 cells were counted per staining condition. n.d.= not detected. Scale bars, 50 μm. See also *Supplementary file 1D*.

The following figure supplements are available for figure 1:

**Figure supplement 1**. Sorted CD133+ cells originate from pancreatic ducts.

---

revealed that the spheres can be passaged up to seven generations (G7, 3 months) and that the total number of cells increased with each generation (*Figure 2C,D*, *Figure 2—figure supplement 1C*). After G7, ductal cell expansion was not achieved, and the spheres were not formed (*Figure 2—figure supplement 1C* and data not shown), supporting the view that ductal epithelial cells are not immortalized, and consistent with the origin of pancreatic cells from donors without neoplasia.

## *Neurog3* converts pancreatic duct cells into progeny expressing islet hormones

The endocrine potential of human or mouse pancreatic ductal cells remains controversial. To investigate the potential of purified human pancreatic ductal cells to achieve an endocrine fate, we used an

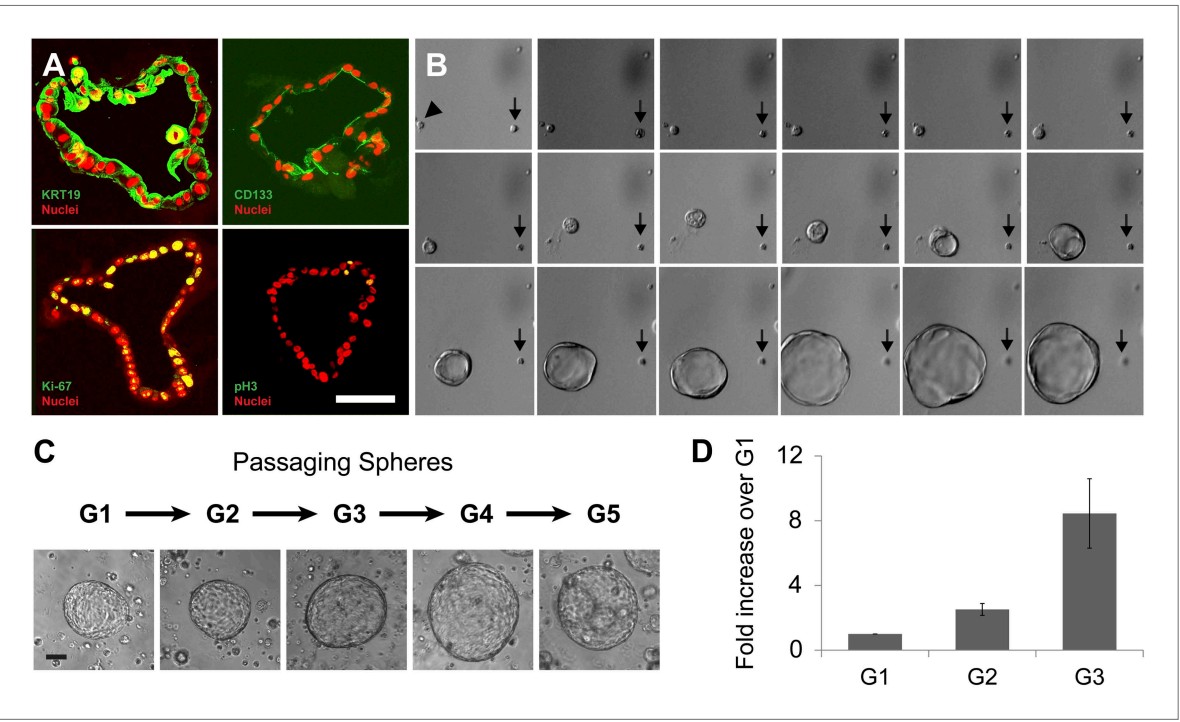

**Figure 2**. Clonal expansion and passaging of ductal spheres. (**A**) Confocal images of 2-week-old spheres immunostained with KRT19, CD133, Ki-67, and Phospho-Histone H3 (all green). Note the apical localization of CD133. Scale bars, 50 μm. (**B**) Representative time-lapse images of sphere formation from single cell (arrowhead). Images taken every 12 hr for 9 days are shown. Arrows point a non-sphere forming cell used as a landmark. (**C**) Representative pictures of spheres after each passage. Scale bars, 100 μm. (**D**) Quantification of cell number in spheres after each indicated passage. Y axis represents fold increase of total cell numbers relative to the one measured in the first 'generation' of spheres (G1).
The following figure supplements are available for figure 2:

**Figure supplement 1**. Quantification of sphere growth and passaging.

adenovirus-mediated transgenic system. *Neurog3* is a transcription factor necessary and sufficient for pancreatic endocrine cell differentiation in vivo (*Gradwohl et al., 2000*; *Gu et al., 2002*) and, combined with other factors, can induce pancreatic acinar-to-islet cell conversion in mice (*Zhou et al., 2008*). To test if *Neurog3* expression could respecify human duct cells toward an endocrine fate, we infected cultured spheres as well as primary CD133+ cells with recombinant adenovirus co-expressing red fluorescent protein and *Neurog3* (Ad-RFP-Neurog3), and assessed changes in gene expression by qRT-PCR (*Figure 3A–C and 4C*). *Neurog3* induced the expression of *NEUROD1*, *INSM1*, and *RFX6* (*Figure 3C*), genes whose mouse homologs are known direct targets of *Neurog3* in pancreas development (*Mellitzer et al., 2006*; *Smith et al., 2010*). Ad-RFP-Neurog3 infection induced expression of the pan-endocrine markers chromogranin A (*CHGA*) and synaptophysin in both primary CD133+ duct cells and cultured spheres (*Figures 3C and 4C*, and data not shown). Ad-RFP-Neurog3 infection also induced expression of mRNA encoding PAX4 and NKX2.2, transcriptional regulators of pancreatic endocrine cell fate (*Sosa-Pineda et al., 1997*; *Sussel et al., 1998*), and mRNA encoding crucial β-cell factors such as the prohormone processing enzymes *PCSK1* (PC1/3) and *PCSK2* (PC2), $K_{ATP}$ channel components *KCNJ11* (KIR6.2) and *ABCC8* (SUR1), and glucokinase (*GCK*) (*Figure 4D*). Moreover, Ad-RFP-Neurog3 significantly induced mRNA encoding the pancreatic hormones ghrelin and somatostatin, but not mRNAs encoding insulin, glucagon, PPY or the intestinal hormones cholecystokinin and gastrin (*Figures 3C and 4D*, *Figure 4—figure supplement 1A*, and data not shown). These findings support the conclusion that human adult pancreatic ductal cells harbor pancreatic endocrine potential upon induction of Neurog3.

Immunostaining confirmed these qRT-PCR findings and demonstrated that only RFP+ cells produced by Ad-RFP-Neurog3 infection were immunostained with antibodies recognizing NEUROD1, NKX2.2, CHGA, SST or GHRL (*Figure 3B,D*, *Figure 3—figure supplement 1*). We also confirmed that no insulin-, glucagon- or PPY-positive cells were observed by immunostaining (data not shown). While only a subset of cells infected with Ad-RFP-Neurog3 (RFP+) expressed CHGA, we noted all GHRL+ or SST+ cells co-expressed CHGA (*Figure 3D*). Quantification of CHGA+ and hormone+ cells revealed that 30% of infected cells (RFP+) expressed CHGA. At least 45% of CHGA+ cells produced SST or GHRL, and less than 2% of CHGA+ cells expressed both hormones (*Figure 3D,E*). Thus, Neurog3 expression efficiently converted primary human ductal cells and cultured ductal epithelial spheres into hormone-expressing cells with cardinal features of endocrine pancreas.

In mice, *Neurog3* gene dosage can determine commitment between exocrine and endocrine lineages in pancreas development (*Wang et al., 2010*). Therefore, we next assessed the possibility that the 70% of RFP+ cells infected by Ad-RFP-Neurog3 failing to express CHGA may have achieved inadequate levels of Neurog3 expression. We fractionated cells produced by Ad-RFP-Neurog3 infection by RFP intensity and measured mRNA expression of Neurog3, CHGA, SST and GHRL by qRT-PCR (*Figure 3F,G*). We found that cell fractions with the highest levels of RFP expression ('P4 and P5', *Figure 3F*) had the highest levels of mouse *Neurog3* mRNA, and only these cell fractions produced mRNA encoding CHGA, SST or GHRL (*Figure 3G*). These data suggest that relatively high threshold levels of Neurog3 may be necessary and sufficient for directing endocrine differentiation of human pancreatic cells.

## Conversion of pancreatic duct cells into progeny that produce, process, and store insulin

The transcription factors *MafA, Neurog3,* and *Pdx1* (a combination hereafter summarized as 'MNP') were sufficient to convert adult mouse acinar cells into insulin-producing cells (IPCs: *Zhou et al., 2008*). We constructed three adenoviruses expressing *MafA, Neurog3,* or *Pdx1* (see 'Materials and methods'; *Figure 4A*), and infected cultured spheres with this MNP combination. Within 5 days after infection, we reproducibly detected *INS* mRNA induction but at extremely low levels relative to adult human islet controls (0.0035 ± 0.0012% of islet levels; *Figure 4B*). Thus, we sought additional factors and discovered that mRNA encoding PAX6, an important regulator of mouse pancreatic endocrine cell development (*Sander et al., 1997*), was induced by MNP to only 0.03% of levels in control islets (*Figure 4—figure supplement 1A*). When combined with *MafA, Neurog3,* and *Pdx1* (encoded in four viruses, '4V'), *Pax6* induced *INS* expression in primary CD133+ ductal cells or spheres by over 30-fold relative to MNP (*Figure 4A,C,D*). We observed ductal conversion to IPCs with four consecutive, independent donors (*INS*, *Figure 4D*). We also detected substantially increased expression of other islet endocrine markers, including *SST, GCK, PCSK1, KCNJ11,* and *ABCC8* (*Figure 4D*).

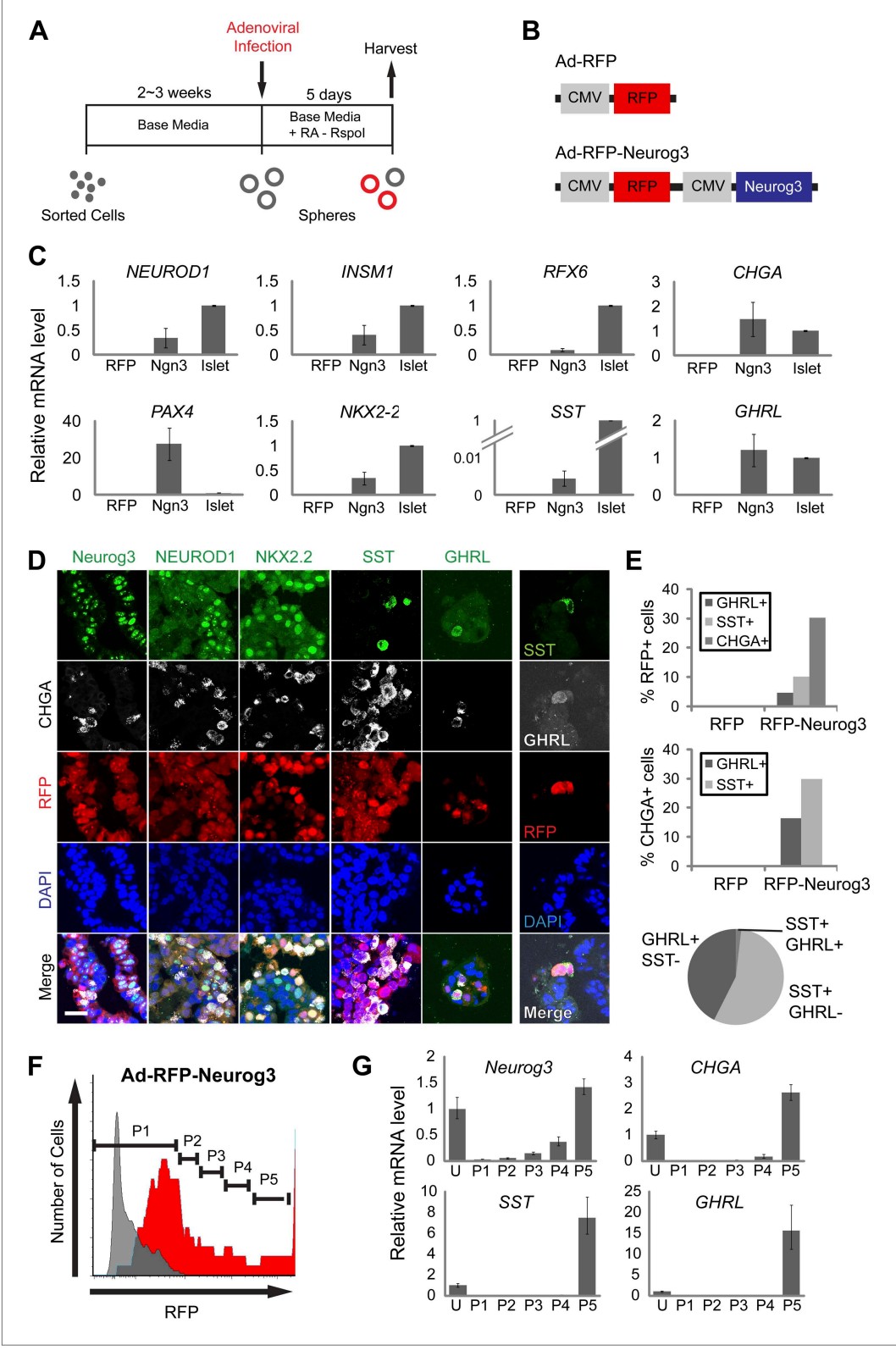

**Figure 3**. Neurog3 is sufficient to convert pancreatic ductal spheres into hormone-expressing endocrine-like cells. (**A**) Schematic of growth and reprogramming strategies. See 'Materials and methods' for details. (**B**) Schematics of adenoviral constructs used. (**C**) Relative mRNA level of Neurog3 targets (*NEUROD1*, *INSM1*, and *RFX6*), endocrine cell-specific genes (*PAX4*, *NKX2.2*, and *CHGA*), and pancreatic hormones (*SST* and *GHRL*). Data are presented as *Figure 3. Continued on next page*

*Figure 3. Continued*

mean ± SEM (n ≥ 3). (**D**) Representative confocal images of Ad-RFP-Neurog3 infected spheres after immunostaining with antibodies specific to mouse Neurog3, NEUROD1, NKX2.2, SST, and GHRL. Note that all hormone-positive cells are CHGA-positive. Right: co-staining of SST and GHRL. Scale bar, 20 µm. (**E**) Quantification of the staining results shown in (**D**). Pie graph represents the percentage of the hormone⁺ cells. (**F**) A representative FACS plot of dissociated ductal spheres infected with Ad-RFP-Neurog3 adenovirus (red) or uninfected control (gray). Fractions P1 through P5 were sorted based on RFP fluorescence intensity. (**G**) qRT-PCR analysis of fractions P1 through P5 from (**F**). 'U' indicates unsorted cells. Analytical duplicates are shown. Data are presented as mean ± SD.
The following figure supplements are available for figure 3:

**Figure supplement 1**. Representative confocal images of spheres infected with control virus (Ad-RFP).

Immunohistochemical analyses demonstrated that the number of Insulin⁺ cells was increased by 18 to 20-fold in spheres transduced by the four factor combination (4V) compared to the MNP combination (*Figure 4F,G*). ELISA studies quantified and confirmed this increase of proinsulin levels, showing that the spheres derived from 4V exposure contained proinsulin levels that averaged 0.7% of those in human islets (*Figure 5E*). Systematic removal of individual factors from this four virus combination revealed that omission of *Neurog3* prevented expression of *INS*, *CHGA* or *SST* (*Figure 4E*). Omission of virus expressing *MafA* or *Pax6* from this combination significantly reduced *INS* expression (*Figure 4E–G*), whereas omission of virus expressing *Pdx1* did not significantly decrease *INS* expression. Thus, *Neurog3*-mediated endocrine cell conversion is required for the production of IPCs as well as other hormone-producing cells from ductal spheres.

Although ELISA studies readily detected proinsulin production by IPCs in our 4V spheres, we failed to detect processed C-peptide by ELISA (*Figure 5E*) or by immunostaining with antibodies recognizing cleaved C-peptide (data not shown). Thus, we sought methods to enhance proinsulin processing in IPCs produced by genetic conversion. For this, we used Ad-*Neurog3*-IRES-eGFP and a second adenovirus constructed to express simultaneously the three transcription factors MAFA, PAX6, and PDX1 (Ad-eGFP-M6P) in cultured G1 spheres (*Figure 5A*, referred to as '4TF' combination). Compared to our standard 5 day post-infection culture (4TF), we found that two additional weeks of culture (referred to as '4TFM') resulted in a 10-fold increase of *INS* mRNA expression in spheres (*Figure 5B,C*, *Figure 5—figure supplement 1A*). We observed conversion to IPCs with five consecutive, independent donors (*INS*, *Figure 5C*), demonstrating the robustness of our conversion method. The total number of converted IPCs appeared unchanged after this extended culture compared to 4V cultures (*Figure 4G*, *Figure 5—figure supplement 1B*), suggesting that *INS* mRNA levels per cell were increased in the 4TFM (4 **t**ranscription **f**actors in two viruses plus **m**aturation period) condition. In addition, mRNA encoding islet amyloid pancreatic polypeptide (IAPP), a β-cell dense core granule component not detectable in standard 4TF conditions, was readily detected in 4TFM spheres (*Figure 5—figure supplement 1A*). Likewise, multiple mRNAs encoding β-cell factors were expressed at levels comparable to those in purified human islets (*Figure 5C*), including the transcription factors *NKX2.2* and *NKX6.1*, *GCK*, glucose transporters *SLC2A1* (GLUT1) and *SLC2A2* (GLUT2), *PCSK1*, *PCSK2*, Zinc transporter *SLC30A8*, *KCNJ11*, *ABCC8*, the voltage-gated calcium channel component *CACNA1C*, regulators of Ca⁺⁺-induced insulin exocytosis like *RAB3A*, *SYT3*, and *VAMP2*, and the postulated maturation marker Urocortin 3 (*UCN3*) (*Suckale and Solimena, 2010*; *Blum et al., 2012*). Immunohistochemical analyses corroborated our qRT-PCR analysis, and showed that converted Insulin⁺ IPCs did not express other islet hormones (*Figure 5D*). Although we were unable to assess endogenous MAFA and PDX1 in cells with virally-expressed exogenous MAFA and PDX1 protein, we readily detected other known β-cell specific markers including NKX6.1, IAPP, and PC1/3 (*Figure 5D*). Moreover, Insulin⁺ cells, but not other hormone⁺ cells, expressed NKX6.1, a transcription factor with expression normally restricted in islets to β-cells (*Figure 5—figure supplement 1C* and data not shown).

To assess enhanced IPC maturation after extended culture (4TFM), we measured proinsulin and insulin C-peptide by ELISA. Total insulin (proinsulin + C-peptide) levels ranged from 3.4 to 15.2 pmol/µg DNA (*Figure 5E*), equal to approximately 9.6% of the total insulin protein level found in human adult islets (*Figure 5E*). Moreover, the percentage of insulin C-peptide processing in IPCs was comparable to that found in adult human islets (IPCs 77–92%; human islets 96–97%), indicating that maturation of

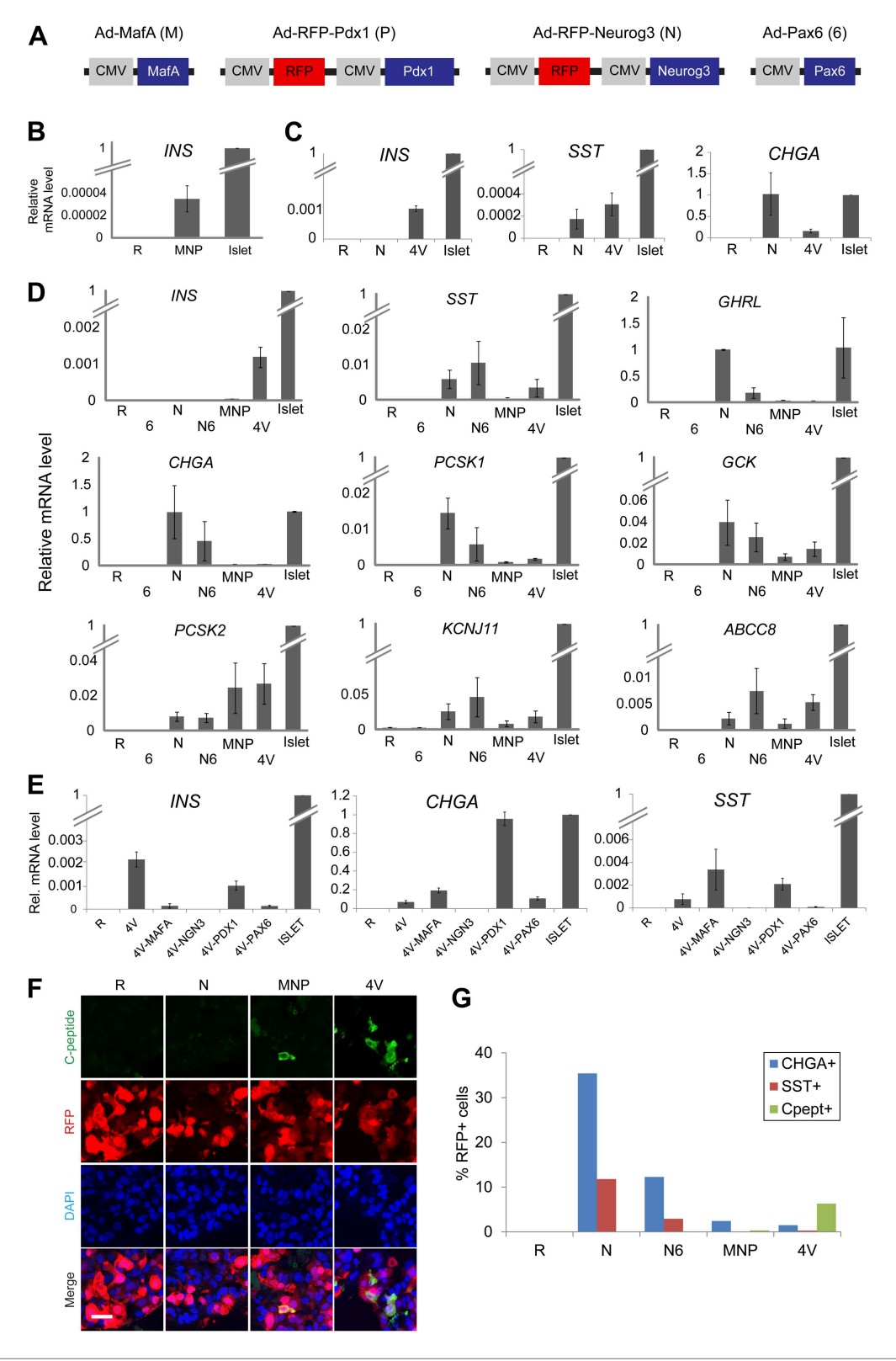

**Figure 4**. Induction of four transcription factors *(Neurog3, MafA, Pdx1, and Pax6)* produces Insulin⁺ endocrine cells in pancreatic ductal spheres in vitro. (**A**) Schematics of adenoviruses used. (**B**) *INS* qRT-PCR analysis of human spheres infected with control (R = RFP) or a combination of MafA (M), Neurog3 (N), and Pdx1 (P) (MNP) n = 4. *Figure 4. Continued on next page*

*Figure 4. Continued*

(**C**) qRT-PCR analysis of *INS*, *SST*, and *CHGA* with freshly sorted CD133+ ductal cells infected with adenoviruses encoding Neurog3 or all four genes (4V) (n = 2). (**D**) qRT-PCR analysis of the spheres infected with a combination of adenoviruses. *Pax6* abbreviated as '6', (n ≥ 3). (**E**) qRT-PCR analysis of the spheres infected with 4V minus each indicated factor n = 2. All bar graph data are presented as mean ± SEM with mRNA levels from purified adult human islets normalized to 1. (**F**) Confocal images of infected spheres after staining with antibodies recognizing C-peptide. Note that adenoviruses encoding *Neurog3* (N) and *Pdx1* (P) also express RFP. Scale bar, 20 µm. (**G**) Quantification of the CHGA-, SST-, and C-peptide-immunoreactive cells in the spheres infected with the indicated combination of adenoviruses. Note that the number of C-peptide-positive cells increased in 4V than MNP by 18–20-fold.

The following figure supplements are available for figure 4:

**Figure supplement 1**. GCG, PPY, and PAX6 mRNA levels after sphere infection with adenovirus combinations.

IPCs during extended culture permitted proinsulin processing (*Figure 5E*). Ultrastructural studies by electron microscopy demonstrated round dense-core vesicles (*Figure 5F*) resembling those in adult human β-cells, including subsets of immature (light core) and mature (dense or crystallized core) vesicles, and vesicles adjacent to the plasma membrane (*Figure 5F*, *Figure 5—figure supplement 1D*). Consistent with the detection of *SST* mRNA (*Figure 5C,D*), we also observed rare cells with irregular electron-dense granules characteristic of islet δ-cells (*Figure 5—figure supplement 1E*; *Klimstra et al., 2007*).

## Regulated insulin C-peptide secretion by IPCs

Native islet β-cells depolarize and secrete insulin and C-peptide in response to glucose and other physiological or pharmacological stimuli, but reconstructing these hallmark functions in progeny of purified primary human non-β-cells has not been previously achieved during in vitro culture. Compared to baseline secretion in media with 0.1 mM glucose, IPCs increased insulin C-peptide secretion by 2.4-fold upon exposure to 2 mM glucose (*Figure 5G*). Similar to insulin release by human islet β-cells (*Lupi et al., 1999*), glucose stimulated the secretion of approximately 4% of total insulin C-peptide in IPCs (*Figure 5—figure supplement 1G*). This effect was blocked when the cells were incubated with glucose and Diazoxide, a drug that opens $K_{ATP}$ channels and prevents glucose-stimulated insulin secretion (*Figure 5G*). However, unlike adult human islet β-cells, the release of insulin by IPCs was not further increased by 11 mM glucose. Islets from fetal or neonatal stages do not show elevated insulin secretion by high level glucose challenge (*Rozzo et al., 2009*), suggesting that IPCs are similar to immature islet β-cells and that further maturation is possible (*Figure 5G*). Calcium and voltage-dependent calcium channels are important regulators of normal insulin secretion after $K_{ATP}$ channel-mediated membrane depolarization in β-cells (*Henquin, 2005*). When calcium was omitted in secretion buffer, C-peptide secretion stimulated by glucose was abolished, but restored upon calcium addition (*Figure 5G*). Insulin C-peptide release by cultured IPCs was also induced by the depolarizing agent potassium chloride (30 mM KCl), an effect reversed by a subsequent wash in media with 4.8 mM potassium ion (*Figure 5G*, *figure 5—figure supplement 1H*). Treatment with tolbutamide, a $K_{ATP}$ channel blocker causing membrane depolarization, also stimulated insulin secretion by IPCs, an effect prevented by omission of calcium (*Figure 5G*). Together with data showing expression of key regulators of stimulus-secretion coupling, these findings provide strong evidence that IPCs produced by conversion and extended culture in our system develop regulated insulin secretion.

We examined the stability of the conversion of human ducts into IPCs by long-term transplantation of the converted spheres into specific transplantation sites of NOD *scid* gamma (NSG) mice (*Figure 5—figure supplement 2*; *Supplementary file 1A*). Human C-peptide was readily detected in kidney grafts harvested at specific times by immunostaining (8/12 cases, *Figure 5—figure supplement 2A*; *Supplementary file 1A*) and by ELISA (9/10 cases, *Supplementary file 1A*) without detectable tumor formation. This also included C-peptide+ IPCs left in the transplant location beyond 5 months (*Figure 5—figure supplement 2A*, d151), suggesting converted IPCs were stable. However, we observed that the total number of grafted C-peptide+ cells was drastically reduced within 2 weeks after transplantation, likely due to the apoptotic cell death. In three independent IPC transplants, however, we were able to detect circulating human insulin in the serum of host mice, and its level increased following

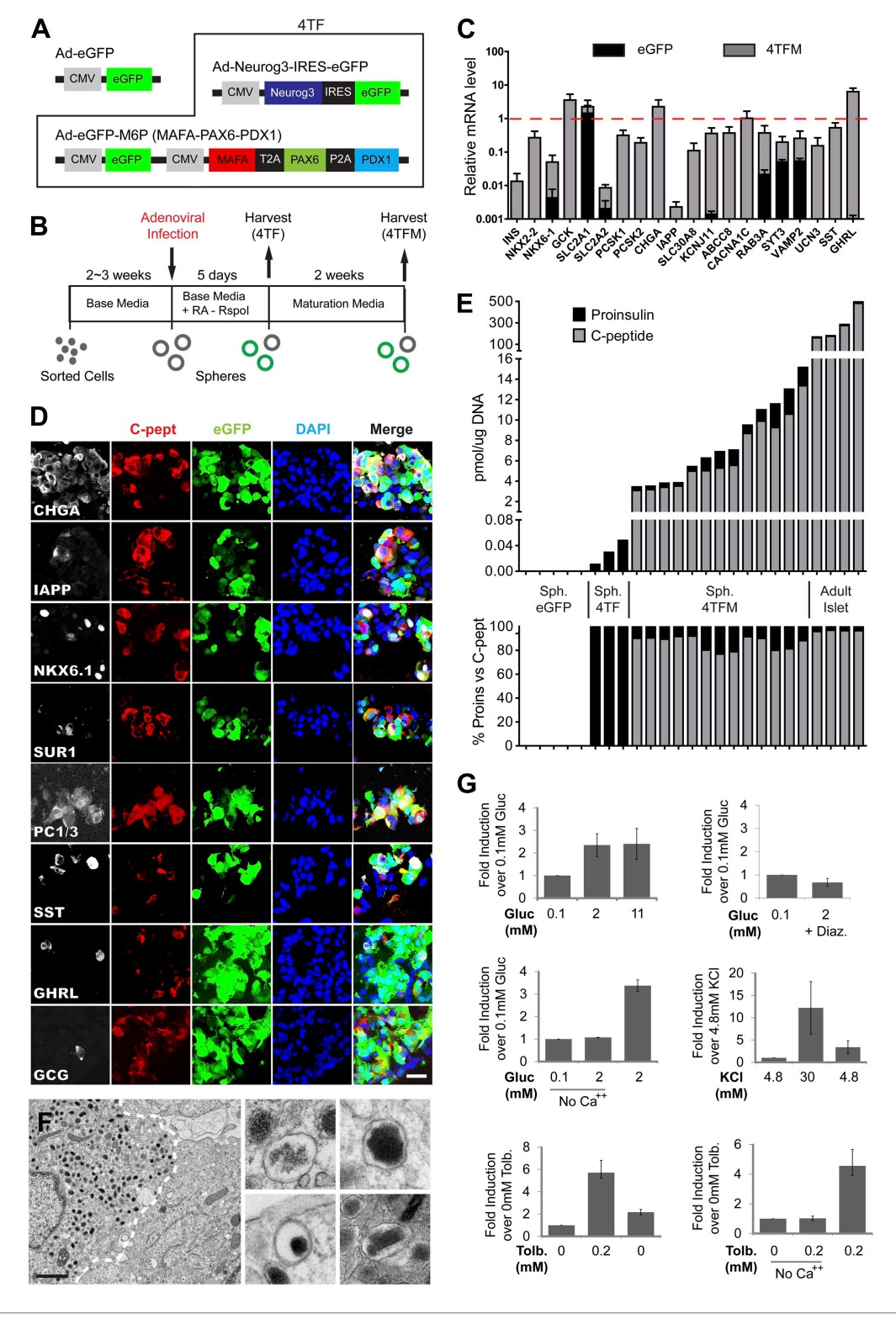

**Figure 5**. Induced insulin-secreting cells resemble functional β-cells. (**A**) Schematic of adenoviral constructs used. See 'Materials and methods' and *Figure 5—figure supplement 5* for details. (**B**) A schematic diagram of growth, conversion, and maturation procedures. (**C**) qRT-PCR analysis of spheres infected with Ad-eGFP (black) or Ad-4TF (gray) followed by extended culture. Data are normalized to adult human islet samples (red dotted line). *Figure 5. Continued on next page*

*Figure 5. Continued*

(**D**) Representative confocal images of 4TFM spheres immunostained with indicated antibodies. Scale bar, 20 µm. (**E**) Quantification of total proinsulin and C-peptide content in GFP, 4TF, 4TFM spheres and human adult islets (Top). Total protein level (pmol) was normalized by total genomic DNA content (µg). Ratio of proinsulin and C-peptide content is presented as % (Bottom). Sph. = Spheres. (**F**) Representative electron microscopic images of 4TFM spheres. Dotted white line demarks cell boundary between converted, granulated (left) and non-converted (right) cells. Dense core vesicles with different morphology in converted cells are shown in the right panels. Scale bar, 1 µm. (**G**) Human C-peptide secretion assay of 4TFM spheres stimulated by the indicated secretagogues and drugs. Gluc = Glucose, Tol = Tolbutamide, Diaz. = Diazoxide. Data are presented as means ± SEM (n = 2 for Diaz.; n ≥ 3 for all other conditions).

The following figure supplements are available for figure 5:

**Figure supplement 1**. Phenotypes of induced Insulin-secreting cells.

**Figure supplement 2**. Grafted IPCs survive long term and secrete insulin C-peptide upon glucose stimulation.

**Figure supplement 3**. Sustained expression of exogenous factors after maturation period.

**Figure supplement 4**. Conversion of human dermal fibroblasts.

**Figure supplement 5**. Protein expression of viral transgenes.

---

intraperitoneal glucose challenge (*Figure 5—figure supplement 2B*; *Supplementary file 1B*). Therefore, these data suggest that despite extensive cell death in early stages of transplantation, IPCs can further mature in vivo and release increased levels of insulin in response to acute glucose challenge.

## Discussion

Methods to regenerate lost or injured cells in diseases like diabetes mellitus are the focus of intensive investigations (*McKnight et al., 2010*; *Benitez et al., 2012*). Generation of insulin-producing cells from human stem cell lines like human ES cells (*D'Amour et al., 2006*; *Kroon et al., 2008*) is an important, and oft-cited 'benchmark', in efforts to achieve β-cell replacement. However, in these prior reports, progeny of human ES cells developed largely as poly-hormonal cells, most frequently expressing both glucagon and insulin. Moreover such hESC progeny failed to secrete insulin in response to glucose or other secretagogues unless transplanted as progenitors for >2 months in mice (*Nostro and Keller, 2012*). This transplant-based maturation strategy was complicated by tumor formation (*Fujikawa et al., 2005*). Thus, it has remained unknown whether human cells can develop solely in vitro to generate glucose-responsive insulin-secreting progeny without tumorigenicity. Our data indicate that in principal this can be achieved, using a small number of genes in sorted human pancreatic ductal cells that convert them toward an islet fate, including progeny that produce, store, and secrete insulin in response to glucose.

Conversion of mouse acinar cells into insulin-producing cells using adenoviral delivery of *Neurog3*, *Pdx1*, and *MafA* was previously reported (*Zhou et al., 2008*). However, it has remained unknown whether human pancreatic cells can be converted using transgenic methods toward a β-cell fate. We were unable to culture and expand primary human pancreatic acinar cells (*Figure 1B* and data not shown); moreover, we found that the combination of these three genes (MNP) was insufficient to reprogram primary or expanded human pancreatic ductal cells toward a β-cell fate, suggesting transgenic conversion may be restricted by species and cell type. Thus, we postulated that additional transcriptional regulators might be needed to promote human ductal conversion toward a β-cell fate. Like Neurog3, MafA, and Pdx1, the transcription factor Pax6 is expressed in both fetal and adult pancreas, and required to achieve appropriately high levels of *Ins* and *Gcg* expression in mouse islet cell development (*Sander et al., 1997*; *Wang et al., 2009*, *2010*; *Pan and Wright, 2011*). Together with the other factors, we found that Pax6 significantly enhanced expression of β-cell markers during ductal reprogramming into β-cells, and was shown as an essential factor for this process. By systematic addition or omission of each transcription factor, we found *PDX1* is not required for IPC formation. Thus, unlike mouse acinar cells (*Zhou et al., 2008*) and human hepatocytes (*Sapir et al., 2005*), human

ductal cells do not require exogenous Pdx1 expression for conversion toward an endocrine fate, for reasons that remain unclear. Our findings are also consistent with recent reports that transgenic adult mouse ductal cells can generate endocrine cells in vivo (*Al-Hasani et al., 2013*).

We initially attempted to induce spontaneous differentiation of pancreatic ductal cells using systematic variations of culture conditions, but these efforts proved unsuccessful (J Lee, unpublished results). During pancreas development, Neurog3 level surges in a subset of pancreatic progenitors located in primitive ducts, inducing development of endocrine cell fates (*Zhou et al., 2007*; *Miyatsuka et al., 2009*). Therefore, based on this model, we attempted to mimic induction of Neurog3 in human ductal cells using adenoviral overexpression of *Neurog3*. We found that Neurog3 was necessary and sufficient for reprogramming human ductal cells, and that the level of ectopic *Neurog3* mRNA expressed in ductal cells correlated well with the extent of endocrine reprogramming, including expression of islet hormones (*Figure 3G*). These findings are reminiscent of studies by Gu et al. showing that reduced *Neurog3* gene dosage in mice leads to respecification of pancreatic endocrine progenitors into ductal and acinar cells (*Wang et al., 2010*). Thus, *Neurog3* functions may be evolutionarily conserved in allocating cells toward an exocrine or endocrine fate (whether in development or experimental cell conversion) in a dosage-dependent manner. Consistent with prior work revealing that Neurog3 attenuates islet cell proliferation (*Miyatsuka et al., 2011*), we did not observe multiple rounds of cell division, an important prerequisite for some de-differentiation events (*Hanna et al., 2009*), during Neurog3-dependent cell conversion. Also, we observed *Neurog3* induction alone can rapidly upregulate endocrine molecular signatures in cultured human ductal cells. Thus endocrine cell conversion described here may involve direct conversion of human ductal cells into endocrine cells, rather than de-differentiation, but additional studies are required to assess this possibility. Our findings, albeit with enforced transcription factor expression in adult cells, indicate that Neurog3 expression is sufficient to induce latent endocrine programs in human adult ductal cells, a capacity not yet clearly demonstrated, to our knowledge.

We demonstrated robust expansion of purified human ductal cells in 3-dimensional culture. The cells were clonally expanded and serially passaged up to seven generations over 3 months, achieving an increase in cell number calculated to be up to 3,200-fold. By contrast, in prior studies, the maximum duration of sustained culture achieved with primary human pancreatic ductal cells was 5 weeks (*Trautmann et al., 1993*; *Bonner-Weir et al., 2000*; *Rescan et al., 2005*; *Hao et al., 2006*; *Yatoh et al., 2007*; *Hoesli et al., 2012*). Moreover, cultured cells in spheres maintained cardinal features of primary pancreatic ducts such as apical-basal polarity and KRT19 expression up to seven generations (*Figure 2—figure supplement 1D,E*). Thus, features of our culture system may be useful for studying the genetics and biology of human ductal cells.

Prior studies have reported that duct-containing fractions from human adult pancreas can form insulin-producing cells in vitro (*Bonner-Weir et al., 2000*; *Hao et al., 2006*; *Heremans et al., 2002*; *Noguchi et al., 2006*; *Koblas et al., 2008*; *Swales et al., 2012*) or after xeno-transplantion in mice (*Yatoh et al., 2007*). However, the possibility of endocrine cell contamination in the initial ductal fraction or feeder/stromal cells used for co-culture was raised by the detection of mRNAs encoding islet cell hormones and other endocrine markers in these and other studies (*Heremans et al., 2002*; *Gao et al., 2005*). Therefore, it remained elusive whether human pancreatic ducts retained the potential to produce islet endocrine cells in adult. In this report, we used FACS to fractionate CD133$^+$ ductal cells and used molecular and immunocytological studies to demonstrate complete elimination of cells expressing markers of differentiated endocrine cells (including islet hormones). Therefore, subsequent conversion of these cells into functional endocrine cells provided unequivocal evidence that endocrine cell-free human adult CD133$^+$ ductal cell fraction can be converted into islet endocrine cells. Centroacinar cells are located at the junction of acini and tip of intercalated ducts (*Cleveland et al., 2012*) and their properties remain poorly understood. These cells express CD133 (*Immervoll et al., 2008*), raising the possibility that our fractionated CD133$^+$ cells also include centroacinar cells. Based on their relative paucity in the pancreas, it is unlikely that centroacinar cells are the exclusive source of spheres within this CD133$^+$ fraction, as more than 11% of CD133$^+$ cells were capable of generating spheres (*Figure 1B*). However, because of difficulties performing lineage-tracing experiments with human samples, we cannot exclude the possibility that centroacinar cells may also contribute to the conversion into endocrine cell lineages.

While expression of *Pax6* along with *Neurog3*, *Pdx1* and *MafA* significantly enhanced expression of *INS* and other β-cell marker genes in converted ductal cells, this transcription factor combination alone

was not sufficient to generate mature IPCs. We found that extending the culture period for 2 weeks after viral infection led to maturation of several hallmark β-cell functions, including expression of key β-cell factors, significant increases of *INS* mRNA and protein levels, proinsulin processing, dense-core granule formation, and Insulin secretion in response to glucose or other depolarizing stimuli. We tested four distinct culture media with or without serum for this extended culture, and all media permitted maturation of these β-cell properties in IPCs (*Figure 5—figure supplement 1F* and see 'Materials and methods'), indicating that the duration of culture is a key variable for promoting β-cell maturation in vitro. After maturation, the spheres contained an average of 7% total insulin compared to human islet controls, and 7–11% of cells comprising these spheres produced insulin C-peptide. Thus, we calculate that each reprogrammed Insulin$^+$ cell produced between 49 and 77% of insulin levels observed in native β-cell controls, a comparable level to the IPCs derived from human ES cells (*D'Amour et al., 2006*).

Is the capacity of human ductal cells to be converted toward endocrine islet fates unique? A prior study by *Sapir et al. (2005)* suggests that human hepatocytes may be induced to express insulin. However, the conversion toward an insulin-producing fate was comparatively poor; resulting cells produced about 10,000-fold lower insulin mRNA level than that of human islets, about 3–4 orders of magnitude lower than from conversion of pancreatic duct spheres. In addition, characteristic dense core vesicles in converted hepatocytes were not observed, indicating insufficient conversion towards β-cells. Here, we also assessed the endocrine potential of primary human dermal fibroblasts, cells successfully 'reprogrammed' toward many non-fibroblast fates, including induced pluripotent stem cells (*Takahashi et al., 2007*), but detected no clear evidence of conversion toward an endocrine or β-cell fate (*Figure 5—figure supplement 4*, see 'Materials and methods' for details). Thus, conversion of human adult duct spheres into cells that produce and secrete insulin is singularly robust. Moreover, unlike prior studies of human ES cells that have high variability among ES cell lines used (*D'Amour et al., 2006*; *Kroon et al., 2008*), we demonstrated conversion toward insulin$^+$ fates by ductal cells from multiple unrelated donors, another feature of the robustness of our methods.

Expression of factors produced from viral transgenes persisted in Insulin$^+$ cells for at least 5 months, evidenced by the GFP expression in transplanted insulin-producing cells (*Figure 5—figure Supplement 2A and 3*). The transgenes delivered by adenovirus do not generally persist in dividing cells (*Zhou et al., 2008*). We speculate that cell cycle arrest in Insulin$^+$ cells may be induced by Neurog3 (*Miyatsuka et al., 2011*), thereby preventing dilution of viral transgene-encoded factors. Thus, further studies are needed to investigate how persistent expression of conversion factors like Neurog3 affects maintenance and maturation of endocrine phenotypes in converted cells. Survival of transplanted insulin-secreting cells produced from ductal cells was poor, and reduced yields following transplantation of ductal cells precluded physiological studies in mouse models of diabetes. Promoting survival of transplanted insulin-secreting cells is a general problem for transplant-based islet replacement approaches. Thus, studies of factors that enhance survival of Insulin$^+$ ductal cell progeny are an important current focus.

In conclusion, our study provides unique evidence that primary human cells can generate progeny that produce, store and secrete insulin in response to glucose or depolarizing agents, the hallmark features of pancreatic β-cells. We also show that human pancreatic exocrine cells, like in mice (*Zhou et al., 2008*), can be converted by transgenes toward an endocrine islet-like cell fate. We speculate that gene-based strategies like those described here may be combined with other methods, including culture modulation by growth factors and small molecules (*Warren et al., 2010*), to optimize endocrine differentiation or conversion of diverse cellular sources to advance cell replacement for diabetes. We speculate that our cell culture system may also serve as the foundation to investigate the genetics and pathogenesis of diverse human diseases rooted in pancreatic ductal cells, including pancreatitis, cystic fibrosis, and adenocarcinoma.

## Materials and methods

### Cell preparation

Institutional review board approval for research use of human tissue was obtained from the Stanford University School of Medicine. Human islet-depleted cell fractions were obtained with appropriate consent from healthy, non-diabetic organ donors deceased due to acute traumatic or anoxic death by overnight shipping from the following facilities: Division of Transplantation (Massachusetts General

Hospital, MA), UAB Islet Resource Facility (University of Alabama at Birmingham, AL), UCSF Diabetes Center (University of California, San Francisco, CA), Kidney/pancreas transplantation center (University of Pennsylvania, PA), Islet Core of the University of Pittsburgh (Pittsburgh, PA), and Human Islet Isolation Program (The Hospital of the University of Virginia, VA). Donor samples with the age range 16–63 years (mean 38.24 years) used for this study are listed in *Table 1*. On receipt, the cell fractions were washed with PBS and cultured with CMRL media (Mediatech, Inc, Manassas, VA) supplemented with 10% heat inactivated fetal bovine serum (FBS, HyClone, Logan, UT), 2 mM GlutaMax (Life Technologies, Grand Island, NY), 2 mM nicotinamide (prepared in PBS, Sigma, St.Louis, MO), and 100 U Penicillin and 100 µg Streptomycin (Pen/Strep, Life Technologies) in a non-coated culture dish at 25.5°C in 5% $CO_2$ until use. For dissociation, the cell pellet was washed with PBS, trypsinized with 0.05% Trypsin-EDTA solution (Life Technologies) for 5 min, and quenched with 5 vol of FACS buffer (10 mM EGTA, 2% FBS in PBS). Cells were collected by centrifugation and further digested in 1 U/ml dispase solution (Life Technologies) containing 0.1 mg/ml DNaseI in PBS on a nutating mixer at 37°C for 30 min. PBS washing was performed after each enzymatic digestion step. After centrifugation, the cell pellet was resuspended in FACS buffer and passed through a 40-µm-cell strainer. Cell viability and number were assessed using a Vi-Cell analyzer (Beckman Coulter, Fullerton, CA) and the samples exceeding 70% cell viability were used for subsequent antibody staining for FACS.

## Cell sorting and culture

Dissociated cells were stained with biotin-conjugated CD133 antibodies (clone AC133 and 293C3, Miltenyi Biotec, Auburn, CA) and then Allophycocyanin-conjugated Streptavidin (eBioscience, San Diego, CA) for 15 min, each at room temperature. Cell pellets were collected by centrifugation and washed with PBS after each staining steps. Propidium Iodide (Life Technologies) staining was used to exclude dead cells. The cells were sorted using a FACSAria II (BD Biosciences, Bedford, MA) and collected in 100% FBS, washed with PBS twice, and resuspended in ice-cold Advanced DMEM/F-12 media (Life Technologies) at a density of 8000 cells/µl. The average percentage of CD133$^+$ fraction was 32.73% (n = 32). 50 µl of growth factor-reduced Matrigel (BD Biosciences) was then added to 30 µl cell suspension and the mixture was placed around the bottom rim of each well. After solidification at 37°C for 60 min, each well was overlaid with 500 µl of modified crypt culture media (*Sato et al., 2009*) comprised of Advanced DMEM/F-12 media supplemented with recombinant human (rh) EGF (50 ng/ml, Sigma), rhR-spondin I (500 ng/ml, R&D systems, Minneapolis, MN), rhFGF10 (50 ng/ml, R&D systems), recombinant mouse Noggin (100 ng/ml, R&D systems), 10 mM Nicotinamide in PBS, and Pen/Strep. The media was changed twice weekly. The spheres were harvested after 2 to 3 weeks for passaging or viral infection. Static and time-lapse images of sphere growth were collected using Zeiss Axiovert 200 inverted microscope and Zeiss Observer.Z1 equipped with a temperature- and $CO_2$-controlled chamber using Axiovision (Carl Zeiss, Germany) and MetaMorph (Molecular Devices, Sunnyvale, CA) softwares, respectively. For harvesting spheres, 500 µl of 2 U/ml dispase (Life Technologies) solution containing 0.1 mg/ml DNaseI in PBS was added in each well and the Matrigel was mechanically disrupted by pipetting and incubated at 37°C for 45 min. The released spheres were collected, washed twice with PBS and used for subsequent applications. For passaging spheres, the harvested spheres were trypsinized at 37°C for 5 min followed by quenching with FBS. The dispersed cells were then used for cell counting with a hemocytometer or were plated as described above.

## Construction of adenoviral vectors

Ad-eGFP and Ad-RFP control adenoviruses were purchased from Vector Biolabs (Philadelphia, PA). Ad-MafA and Ad-Neurog3-IRES-eGFP were described previously (*Tashiro et al., 1999*). To construct Ad-RFP-Neurog3 and Ad-RFP-Pdx1 adenoviruses, mouse cDNAs for Neurog3 (BC104326) and Pdx1 (BC103581) were purchased from Open Biosystems (Lafayette, CO) and the inserts were obtained by restriction enzyme digestion with EcoR V/BamH I and EcoR V/Msc I, respectively. The inserts were then subcloned into multiple cloning sites of Dual-RFP-CCM shuttle vector (Vector Biolabs) and adenoviruses were constructed by Vector Biolabs. For Ad-eGFP-M6P, human *MAFA* cDNA (gift from M German), *PDX1* (NM_000209; GeneCopoeia, Rockville, MD), and *PAX6* (BC011953; Open Biosystems) were used for PCR amplification with the primers shown in *Supplementary file 1C* to add T2A, P2A, restriction enzyme sites, and/or tagging proteins (*Figure 5—figure supplement 5*). A fused construct of MAFA-T2A-PAX6 was generated by PCR with *MAFA* and *PAX6* PCR amplicons as templates. Similarly, PCR products for PAX6 and PDX1 were used to construct PAX6-P2A-PDX1. Next, MAFA-T2A-PAX6,

PAX6-P2A-PDX1, and pDual-GFP-CCM vector (Vector Biolabs) were cut with BglII/PstI, PstI/EcoRI, and BglII/EcoRI, respectively, and ligated with NEB quick ligation kit (New England Biolabs, Ipswich, MA) followed by transformation of TOP10 chemically competent cells (Invitrogen, Carlsbad, CA). The construct was then used for generating adenoviruses by Vector Biolabs.

## Sphere infection and post-infection culture

Spheres were infected at 37°C in suspension overnight at a multiplicity of infection (MOI) 100 for Ad-MafA and Ad-eGFP-M6P, or MOI 500 for the rest of viruses used. The spheres were then washed twice with culture medium and embedded in Matrigel as described above. The infected spheres were overlayed with sphere growth media without R-spondin I and with 0.33 µM all-trans retinoic acid (Sigma), and cultured for 5 days. For extended culture, the media was replaced with either (1) DMEM with high glucose (Life Technologies) supplemented with 10% FBS (Hyclone) and Pen/Strep (Life Technologies) for 2 weeks (referred as 'DF' in *Figure 3—figure supplement 1F*), (2) DF plus 20 mM KCl and 10 µM R0-28-1675 (glucokinase activator; Axon Ligands) for 2 weeks (referred as 'DFK'), (3) DF for one week and then DMEM/F-12 media (Life Technologies) supplemented with 0.5 × N2 supplement (Life technologies), 0.5 × B27 (Life technologies), 0.2% BSA (Sigma), 1% ITS supplement (Life Technologies), 10 mM nicotinamide, 10 ng/ml recombinant human basic FGF (R&D systems), 50 ng/ml Exendin-4 (R&D systems), recombinant human BMP-4 (R&D systems) for additional 1 week (referred as 'Z'; *Zhang et al., 2009*), or (4) DMEM high glucose supplemented with 1 × B27, 55 nM GLP-1, 50 ng FGF10 (R&D Systems), and Pen/Strep for 3 days followed by 5 days with DMEM high glucose supplemented with 1 × B27, 55 nM GLP-1 (Sigma), 10 µM DAPT (Sigma), and Pen/Strep, then for 6 days with CMRL1066 media (Mediatech) supplemented with 1 × B27, 55 nM GLP-1, 50 ng HGF (R&D Systems), 50 ng IGF-1 (R&D Systems), and Pen/Strep (referred as 'T'; *Thatava et al., 2011*). The media was replaced every other day unless otherwise noted.

## cDNA preparation and qRT-PCR analyses

Total RNA was prepared from sorted cells or cultured spheres with QIAGEN RNeasy micro kit (QIAGEN Sciences, MD), and used for cDNA synthesis using QIAGEN Omniscript RT kit (QIAGEN), according to the manufacturer's protocol. Relative mRNA level was measured by qRT-PCR of each cDNA in duplicate with gene-specific probe sets (Applied Biosystems, Foster City, CA) with TaqMan Universal PCR Master Mix (Applied Biosystems) and the ABI Prism 7500 detection system (Applied Biosystems). Normalizations across samples were performed using β-actin primers. Information of the primer and probe sets is available upon request.

## Immunohistochemistry

For immunohistochemical analyses, cultured spheres were harvested, washed with PBS, mixed with 20 µl of Collagen Gel Kit (Nitta Gelatin, Osaka, Japan), solidified at 37°C for 1 hr, fixed with 4% para-formaldehyde for 2 hr at 4°C, cryoprotected in 30% sucrose solution in PBS overnight, embedded in OCT on dry ice, and sectioned in 8 µm thickness. For sorted cells, the cell suspension was washed once and resuspended with 20 µl of PBS, placed on a Polysine slide (Thermo scientific, Waltham, MA), and waited for 30 min at room temperature (RT) to let the cells sit on the slide glass by gravity. Then the solution was removed carefully and 40 µl of 4% paraformaldehyde was added. After 10 min of incubation at RT, the fixative was removed and the slides were washed with PBS three times for 5 min each. After removal of PBS, the slides were dried at RT for 1 hr and stored at −20°C. For immunostaining transplanted IPCs, grafted organs (kidney, EFP, or liver) were harvested, fixed with 4% paraformaldehyde overnight at 4°C, cryoprotected in 30% sucrose solution in PBS overnight, embedded in OCT on dry ice, and sectioned in 8 µm (kidney and liver) or 40 µm (EFP) thickness. The primary antibodies used were rabbit anti-Amylase (1:1000; Sigma), goat anti-Amylase (sc-12821; 1:200; Santa Cruz Biotechnology, Dallas, TX), CD133 (1:100 each; clone AC133 and 293C3; Miltenyi Biotec, Auburn, CA), rabbit anti-ChromograninA (20085; 1:100; Immunostar, Hudson, WI), mouse anti-ChromograninA (LK2H10; 1:200; Cell Marque, Rocklin, CA), mouse anti-CK19 (KRT19) (M0888; 1:200; DAKO, Carpinteria, CA), rabbit anti-CK19 (319R-15; 1:200; Cell Marque), rabbit anti-CPA1 (1810-0006; 1:100; AbD Serotec, UK), rabbit anti-C-peptide (#4593B; 1:200; Cell Signaling Technology, Danvers, MA), mouse anti-C-peptide (capt) (1:100; Mercodia, Sweden), mouse anti-Flag (F1804; 1:1000; Sigma), goat anti-GHRL (sc-10368; 1:200; Santa Cruz Biotechnology), guinea pig anti-Glucagon (4031-01; 1:200; Linco, Billerica, MA), mouse anti-HA (MMS-101P-1000; 1:1000; Covance), mouse anti-HuNu (MAB1281; 1:200; Millipore, Billerica,

MA), mouse anti-IAPP (MCA1126T; 1:200; AbD serotec), rabbit anti-Ki-67 (NCL-Ki67p; 1:100, Leica Microsystems, Germany), rabbit anti-Myc (sc-789; 1:1000; Santa Cruz Biotechnology), mouse anti-NeuroD (sc-46684; 1:10; Santa Cruz Biotechnology), mouse anti-Neurog3 (F25A1B3; 1:4000; DSHB, Iowa City, IA), mouse anti-Nkx2.2 (74.5A5; 1:10; DSHB), mouse anti-Nkx6.1 (F55A10; 1:200; DSHB), rabbit anti-PC1/3 (PCSK1, AB10553; 1:200; Millipore), rabbit anti-phospho-H3 (06-570; 1:500; Millipore), goat anti-PPY (NB100-1793; 1:200; Novus Biologicals, Littleton, CO), rabbit anti-Somatostatin (1:200, DAKO), goat anti-Somatostatin (sc-7819; 1:200; Santa Cruz Biotechnology), goat anti-SUR-1 (sc-5789; 1:50; Santa Cruz Biotechnology). Tyramide signal amplification (Perkin Elmer, Waltham, MA) was used for antibodies against Neurog3, NeuroD, Nkx2.2, Nkx6.1, and PC1/3. Antigen unmasking (H-3300; Antigen Unmasking Solution, Citric Acid Based, Vector Laboratories, Burlingame, CA) was performed for anti-Flag antibody staining. The Neurog3, Nkx2.2, and Nkx6.1 antibodies developed by Dr OD Madsen were obtained from the Developmental Studies Hybridoma Bank (DSHB) developed under the auspices of the NICHD and maintained by The University of Iowa, Department of Biological Sciences, Iowa City, IA 52242. Secondary antibodies used were from Jackson ImmunoResearch (West Grove, PA) or Molecular Probes (Eugene, OR). Stained sections were mounted with VECTASHIELD Mounting Medium with DAPI (Vector Laboratories). Fluorescence images were taken using Zeiss Axio Imager.M1 or Leica SP2 inverted confocal laser scanning microscope.

## Electron microscopy

The samples were fixed in Karnovsky's fixative: 2% Glutaraldehyde (EMS Cat# 16000) and 4% Paraformaldehyde (EMS; Electron Microscopy Sciences, Hatfield, PA) in 0.1 M Sodium Cacodylate (EMS) pH 7.4 for 1 hr at RT then cut, post fixed in 1% Osmium tetroxide (EMS) for 1 hr at RT, washed three times with ultrafiltered water, then en bloc stained for 2 hr at RT or moved to 4°C overnight. The samples were then dehydrated in a series of ethanol washes for 15 min each at 4°C beginning at 50%, 70%, 95%, where the samples are then allowed to rise to RT, changed to 100% two times, followed by Acetonitrile for 15 min. The samples are infiltrated with EMbed-812 resin (EMS) mixed 1:1 with Acetonitrile for 2 hr followed by two parts EMbed-812 to 1 part Acetonitrile for 2 hr. The samples were then placed into EMbed-812 for 2 hr and then placed into molds, and resin filled gelatin capsules with labels were orientated over the cells of interest and placed into 65°C oven overnight. Sections were taken between 75 and 90 nm on a Leica Ultracut S (Leica, Wetzlar, Germany), picked up on formvar/Carbon coated slot grids (EMS Cat#FCF2010-Cu) or 100 mesh Cu grids (EMS). Grids were contrast stained for 15 min in 1:1 saturated UrAcetate (~7.7%) to 100% ethanol followed by staining in 0.2% lead citrate for 3 to 4 min. JEOL JEM-1400 TEM was used to observe at 120 kV and photos were taken using a Gatan Orius digital camera.

## C-peptide secretion and content measurement

C-peptide secretion assay and content measurement were performed as described previously with minor modification (*Chen et al., 2001*). Briefly for secretion assay, media was replaced a day before assay was performed. On the day, each well with matrigel-embedded spheres was incubated with fresh media for 2 hr, washed twice with plain Krebs-Ringer bicarbonate buffer (KRBB), and incubated twice with plain KRBB for 1 hr each for thorough washing. Next, the spheres were incubated consecutively with 400 µl KRBB containing indicated concentrations of glucose (Sigma) with or without 0.5 mM Diazoxide (Sigma), KCl (30 mM, Sigma), or Tolbutamide (0.2 mM, Sigma) for 2 hr each. KRBB without Calcium (No Ca$^{++}$) was prepared by omission of CaCl$_2$ and addition of 1 mM EGTA (Sigma). Secreted C-peptide level was measured with Human Ultrasensitive C-peptide ELISA kit (Mercodia). For C-peptide content measurement, the spheres were harvested in 1.5 ml microfuge tube, washed with PBS, resuspended with 300 µl of ice-cold TE/BSA buffer (10 mM Tris-HCl, 1 mM EDTA, 0.1% wt/vol BSA, pH 7.0), and sonicated with Bioruptor Sonicator (Diagenode, Denville, NJ). Half of the lysate was used for genomic DNA isolation and quantification with Quant-iT PicoGreen dsDNA Assay Kit (Invitrogen). Same volume of acid alcohol (75% vol/vol ethanol, 2% vol/vol concentrated HCl, 23% vol/vol H$_2$O) was added to the rest of lysate to extract C-peptide by rocking overnight at 4°C. The extract was then neutralized with 10 vol of PBS and used for C-peptide ELISA.

## Transplantation

Transplantation in kidney capsule, epididymal fat pad (EFP), or in the liver by portal vein injection was performed as previously described (*Kroon et al., 2008*; *Alipio et al., 2010*; *Wang et al., 2011*). For transplantation in kidney or EFP, converted spheres with or without extended culture were harvested

and mixed with or without mouse embryonic fibroblasts (*Supplementary file 1A*). The spheres were then mixed with matrigel to make a final volume of 10 µl for kidney transplantation or overlayed on pre-wet gelfoam for EFP transplantation. For liver transplantation, single cells produced by trypsinization of harvested spheres were resuspended in 100 µl PBS and injected into the portal vein with a 27 G needle. All animal experiments and methods were approved by the Institutional Animal Care and Use Committee (IACUC) of Stanford University.

### In vivo glucose-stimulated insulin secretion assay

Secretion of human Insulin or C-peptide by glucose injection was measured as previously described (*Kroon et al., 2008*). Briefly, transplanted mice were fasted overnight (14–16 hr) and 120 µl of blood was collected from tail into Microvette CB300LH (Sarstedt, Germany) to prepare 50 µl of serum. 3 g/kg glucose was then injected and blood was collected again 30 min after glucose administration. Secreted C-peptide or insulin level was measured with Human Ultrasensitive C-peptide or Insulin ELISA kits (Mercodia).

### Human adult dermal fibroblast culture and conversion assay

Human adult dermal fibroblasts (Coriell Institute for Medical Research, Camden, New Jersey, USA) were cultured and maintained as described previously (*Yoo et al., 2011*). The cells were either trypsinized for suspension infection (as was described above for ductal spheres) or infected as adherent cells in six-well plates by direct addition of virus into the culture medium, with Ad-eGFP (GFP) or Ad-eGFP-M6P and Ad-Neurog3-IRES-eGFP (4TFM). The same MOIs used for ductal sphere infection were also used. The suspension-infected cells were harvested the following day and embedded in Matrigel as described above for infected ductal spheres. The culture was maintained for additional 18 days to match the duration of infected ductal sphere maturation. The infected adherent cells were cultured with virus for 48 hr and the media was replaced. The culture was maintained for additional 10 days, passaged in 1:3 ratio due to confluency, re-plated, and cultured additional 7 days to match the duration of infected ductal sphere maturation. In both cases, media was replaced every other day. Three independent experiments were performed for both conditions and each experiment at least in duplicates. RNA isolation, cDNA preparation, and qRT-PCR were performed with primers specific to human *INS*, *CHGA*, and β-actin as described above.

## Acknowledgements

We thank Drs H-E Hohmeier and Christopher Newgard (Duke University) for helpful discussions and advice, Dr M German (UCSF) for human *MAFA* cDNA, Drs K D'Amour and E Kroon (Viacyte) for advice on EFP transplantation, Ms S Bryant (University of Alabama), Drs A Naji and C Liu (University of Pennsylvania), Dr X Huang (University of Virginia) for human non-islet cell processing, Mr J Perrino and the Stanford Cell Sciences Imaging Facility for transmission electron microscopy and confocal microscopy, Ms E Snyder and Mr J Albright for technical support, and members of the Kim Laboratory for comments on the manuscript.

## Additional information

### Funding

| Funder | Grant reference number | Author |
|---|---|---|
| JDRF | 43-2010-347 | Jonghyeob Lee, Jing Wang, Seung K Kim |
| Howard Hughes Medical Institute | | Seung K Kim |
| Larry L Hillblom Foundation | 2008-D-018-FEL | Jonghyeob Lee |

The funders had no role in study design, data collection and interpretation, or the decision to submit the work for publication.

### Author contributions

JL, Conception and design, Acquisition of data, Analysis and interpretation of data, Drafting or revising the article; TS, Contributed initial conception and execution of pancreatic sphere culture; YL, Contributed

cell sorting and ductal sphere culture; JW, Renal subcapsular transplantation and human tissue assessment; XG, Contributed immunohistochemical and morphometric studies; JL, JFM, GLS, RB, Procurement of cadaveric human pancreas and islet-depleted cell preparation; SM, J-iM, Provision of Ad-MafA and Ad-Neurog3-IRES-eGFP; SKK, Conception and design, Analysis and interpretation of data, Drafting or revising the article

## Ethics

Animal experimentation: This study was performed in strict accordance with the recommendations in the Guide for the Care and Use of Laboratory Animals of the National Institutes of Health. All of the animals were handled according to approved institutional animal care and use committee (IACUC) protocols (#10160) of the Stanford University. All surgery was performed under anesthesia, and every effort was made to minimize suffering.

## Additional files

### Supplementary files

• Supplementary file 1. (**A**) Summary table of converted sphere transplantation. (**B**) Glucose-stimulated insulin C-peptide secretion in vivo. (**C**) PCR primers used for Ad-GFP-M6P construction. (**D**) Quantification data of cell immunostaining after FACS (*Figure 1F*).

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
