## [Decision Letter]

[Editors’ note: this article was originally rejected after discussions between the reviewers, but the authors were invited to resubmit after an appeal against the decision.]

Thank you for choosing to send your work entitled “Expansion and Conversion of Human Pancreatic Ductal Cells into Insulin-Secreting Endocrine Cells” for consideration at *eLife*. Your full submission has been peer reviewed by one of our Senior editors, Janet Rossant, and two other reviewers, and the decision was reached after discussions between the reviewers. We regret to inform you that your work will not be considered further for publication at this point.

The reviewers and the Senior editor have had an extensive online discussion about your paper, after exchanging the reviews. While they all feel that the experiments are carefully carried out, and the data presented are robust, in the end they were not convinced that the study as a whole provided a major step forward in the drive towards generating functional beta cells from other cell types. It was noted that you have not demonstrated whether the use of ductal cells (CD133^+^ cells) is advantageous over the use of other cell types for transdifferentiation. What would happen if the same factors were used to reprogram other cell types, even non-pancreatic cells, such as fibroblasts? The fact that ductal cells can respond to exogenous transcription factors does not directly demonstrate that these cells have latent potential to form beta cells, as claimed in the Abstract. It was also noted that the beta cells produced are not apparently fully mature and, therefore, the long-term significance of this approach for human therapy is unclear. The ability to grow human ductal cells in vitro is interesting and a further analysis of the non-transduced cells to respond to external signals and undergo differentiation into different cell types would be interesting.

Given these major concerns and the amount of extra work that would be needed to address them, the decision is to reject the manuscript at this time. A majorly enhanced experimental study including assessing whether ductal cells are uniquely responsive to these inducing factors, better characterization of the cells produced, and a further analysis of the unmanipulated ductal cells could form the basis of a new submission at a later date. The major points from the full reviews are provided below.

*Reviewer #1*:

In this manuscript the authors show that they can sort human cadaveric pancreas tissue with antibody to CD133 and that this enriches for pancreatic ductal epithelium. They then show that they can generate clonal spheres from these cells that can be passaged several generations in culture. They then infect these cultures with adenovirus expressing first neurogenin and then additional sets of beta cell inducing transcription factors and show that they can induce expression of endocrine phenotypes in the spheres and that a combination of 4 factors generates insulin-producing cells that show some degree of glucose regulation. The final most successful converted cells express 7% of the levels of insulin found in adult islets, but when transplanted to the kidney capsule in mice, they were able to detect some human insulin in serum. However, in these grafts cell survival was poor, so they were unable to test the ability of the cells to rescue diabetic mice.

This study is well performed and does indicate that ductal cells may be responsive to exogenous transcription factors that can drive towards an islet cell fate.

Major comments:

1) It is not clear whether conversion to islet cells is a unique property of ductal cells or whether other pancreatic cells or other cell types could respond in the same way. Other groups have suggested that hepatocytes can be transdifferentiated to insulin-producing cells- is this system more or less effective?

2) The final cocktail of transcription factors is stated to produce monohormonal insulin-producing cells, but this is not directly shown in the figures. This is an important point because many other differentiation assays generate fetal-like polyhormonal cells that cannot respond to glucose in the manner of adult beta cells. The cells produced here are not fully functional adult-type cells.

3) How long does expression of the exogenous factors persist? Is it required for ongoing maintenance of the cells or can you demonstrate independence of the exogenous factors?

4) How sure are they that the starting population is pure ductal cells, given that CD133 is not exclusively expressed in ductal epithelium in the pancreas? Can they double sort with a general epithelial marker to further purify the population, given that CD133 only enriches 4-fold for sphere-forming cells?

*Reviewer #2*:

In this study the authors describe a method to isolate and expand human ductal cells using an antibody against CD133. Using culture conditions similar to those described by Dr. Hans Clevers (Sato et al.), they were able to culture CD133^+^ cells as epithelial spheres that maintain a ductal phenotype and lack acinar and endocrine markers. Furthermore, the authors were able to reprogram the CD133-enriched population to endocrine cells by infecting isolated CD133^+^ cells and/or CD133^+^ -derived spheres with adenoviruses expressing ngn3, MAFA, PAX6 and PDX1. On average the authors are able to generate 10% insulin+ cells that resemble fetal beta cells, as they secrete insulin in response to low level of glucose (2mM), but fail to respond to higher glucose concentration (11mM). Following transplantation of the reprogrammed spheres into NSG mice, they observed that survival of the graft after transplantation was poor. However, they were able to detect human insulin in the serum of the host mice and showed that insulin levels increased after glucose challenge, suggesting in vivo maturation, albeit few mice were analysed. In general this work is very well done, with convincing images and clear data. There are only some minor points that need to be clarified.

Minor comments:

1) As CD133 is detected in centroacinar cells as well as in ductal cells (Immervoll et al JHC 2011), the authors should include additional acinar markers in their expression profile (Figure 1) to exclude acinar contamination.

2) Please include the average percentage of CD133^+^ cells detected in human pancreas and the purity of the sorted populations.

3) Figure 2 please include co-staining of KRT19 and CD133.

4) The authors state “Time-lapse imaging revealed that spheres arose from single isolated CD133^+^ ductal cells”. However, this statement is not accurate unless the purity of the sort was 100%.

5) Does the percentage of CD133^+^ cells decrease in culture? What is the percentage of CD133 after 3 months in culture?

6) The authors state that Insulin^+^ IPCs did not express other islet hormones, the authors should include co-staining of c-peptide with GCG, PP and Ghrelin in Figure 5.

*Reviewer #3*:

The claim that adult human pancreatic duct cells have a latent capacity for endocrine differentiation is correct, but only in this context of extremely strong transcription-factor-based enforced differentiation. I always wonder what MNP6 would do to a non-pancreatic cell type, and therefore if this effect is a specific latency of pancreatic duct cells or not (the paragraph starting ‘The transcription factors *MafA*, *Neurog3* and *Pdx1*…’ is more important as a result if there is something these cells can do that is not ever seen with the 4-factor combination MNP6 (4TF) or 4TFM).

Essential is the claim that the insulin-producing cells are mono-hormone-positive, but this is not shown in the paper. The authors refer to the Figure 5 as the one showing no double-positives, but no Gcg, PP, or Ghrl are shown here.

Some more clarity on this aspect seems critical. Assays for Gcg and other hormones are either referred to as data not shown, or this aspect is stated but the figure does not have the supporting data. Gcg immunodetection was done on cells from N alone? Gcg, PP were tested on the 4-factor combinations?

Some explanation of why spontaneous (that is, non-TF-enforced) differentiation to endocrine fates was ruled out.

Is Pax6 already expressed in the MNP-adenovirus cultures? If so, why is more needed?

The part on “We sought new methods to mature the cells” (my words) reads strangely. It's the same method, just extending the time frame, I believe, and I would simplify this text. Another comment here is that we revert to the MNP6 mixture (4TF), having just been told that Pdx1 can be removed without impact - why?

Figure 2—figure supplement 1 has an incorrect y-axis. 10,000 percent to the log(base10) is 4, correct. This graph needs altering – why not just “fold” for cell number? Related to this, Discussion asserts 3,000-fold, but this is just once? Up to 3,000-fold, and more typically xx-fold? Why do the cultures suddenly go bad at G7 (see text) – what happens – sudden apoptosis? #48 seems to be continuing even at G7 – please amend this text.

Does the 3-factor system (MNP) work in the authors' hands on mouse acini? And/or duct?

Does the CD133 separation method include centroacinar cells (CAC) or not? What is the photograph in Figure 1 – ‘tip’ of duct dives out of plane of section before it is reached, and therefore the CAC cannot be seen in this panel? CAC could have a specific latency not seen in duct cells.

The idea that lineage-tracing methods are hard to apply in human cells should be stated, as everything else hinges on numerical arguments on cleanliness of cell separations, etc.

chgA is an endocrine differentiation marker. This statement is meant to indicate that full-blown differentiation to the hormone-expressing state requires a substantial pulse of Ngn3. What about other hormones, even indicating non-pancreatic cell types?

A major point is the longevity of the pulse of Ngn3 and the other factors achieved with these methods, and the detection of a program that runs from the endogenous loci with or without the continuous presence of N, MNP, MNP6 (4TF, 4TFM). Would this method pulse the cells or not, and is continuous presence of some of the factors preventing their full differentiation to the most mature state?

Title of the section starting ‘Although ELISA studies readily detected Proinsulin production by IPCs…’ reads odd to me: ‘genetic conversion’ reads as if the genome of the adult duct cells is being altered in some manner.

Systematic removal of factors from MNP6 mixtures: Why can Pdx1 be removed without any impact?

Various ‘obvious’ markers were not tested, or the manuscript is incorrect in not showing such ‘easily pointed out’ questions. Pdx1 is produced, by immunofluorescence assay, within these induced beta cells? To normal levels? MafA/B, etc? Nkx6.1 was assessed, but the ‘dogmatic’ mature β-cell TF list should be addressed, at least.

It does seem difficult to follow 4V, 4TF, 4TFM, MNP. Seems as if there is a mixed descriptor used for the same manipulation, at least sometimes; I suggest simply checking for a way of making the text fully consistent throughout.

---

## [Author Response]

We are grateful that the initial review has provided such useful suggestions for improving our study, and that the overall view of the experimental strategy, concepts, and data quality were so positive. We have provided experimental data, most of it new, to address all the remaining major concerns summarized in the decision letter, including (1) new data with human fibroblasts to assess whether primary human pancreatic ductal cells are uniquely responsive to conversion conditions identified here, (2) better characterization of the cells produced, especially studies to establish their mono-hormonal development, and (3) further analysis of unmanipulated ductal cells. We trust our responses to these requests for additional data meet or exceed the reviewers' expectations.

In addition to the detailed responses to specific points below, we would like to address the general point about the importance of the findings presented here. We thank the reviewers for the opportunity to clarify why our work “represents a major step forward in the drive toward generating functional beta cells from other cell types”.

An important, and oft-cited ‘benchmark’, in this effort is production of insulin-producing cells from stem cell lines like human ES cells (D’Amour et al., 2006; [28] and subsequent work). However, these cells develop largely as poly-hormonal cells (most frequently expressing both glucagon and insulin) and fail to secrete insulin in response to glucose or other secretagogues unless transplanted for >2 months in mice (reviewed in [38]). Thus, it has remained unknown whether human cells can develop solely in vitro to generate glucose-responsive insulin-secreting progeny and without tumorigenicity. Our data indicate that in principle this can be achieved. Moreover, the depth and quality of phenotyping we perform to characterize achievement of beta-cell fate (in our view) is unmatched by any other prior study, with inclusion of ultrastructural, in vitro, and in vivo secretion data.

A second as yet unanswered question in the field is whether human pancreatic cells can be converted using transgenic methods toward a ?-cell fate. The Melton group (67) established that this was possible in mice using viral methods in exocrine acinar cells, but since that seminal study the field has yearned to know how relevant this strategy might be for human cells, especially human pancreatic cells. We feel that our work unequivocally establishes that human pancreatic ductal exocrine cells have latent potential to produce functional endocrine islet-like cells.

Third, there is a longstanding debate about the ability of pancreatic ductal cells to generate endocrine progeny, with the majority of experimental assessment of this important question previously performed in transgenic mice. Thus, we feel our work provides important unique evidence that pancreatic ductal (or centroacinar) cells have such endocrine potential. Moreover, our work addresses a gap in knowledge in our field about (and a new system for evaluating) the potential of human ductal cells for alternate fates.

The identification of a new cellular source and genetic strategy for generating progeny with features of functional ?-cells should advance the long-term development of cell replacement options in type 1 diabetes. Thus, we feel quite strongly that our work provides a major contribution by addressing and answering multiple outstanding questions in diabetes research and pancreas biology. We hope that our revision clarifies this contribution and is now deemed worthy of publication in *eLife.*

Reviewer #1:

*1) It is not clear whether conversion to islet cells is a unique property of ductal cells or whether other pancreatic cells or other cell types could respond in the same way. Other groups have suggested that hepatocytes can be transdifferentiated to insulin-producing cells – is this system more or less effective*?

We fully agree with this reviewer and the similar comment from Reviewer #3 that additional assessment of other human cells would enhance the impact and conclusions from our study. Thus, we attempted to convert human adult dermal fibroblasts using our conversion methods; these cells have been previously used successfully in other reprogramming experiments (65). With the same 4TFM condition and Insulin or ChromograninA (ChgA) mRNA levels as readouts, we detected no clear evidence of conversion toward an endocrine or β-cell fate (n ≥ 6 from 3 independent experiments: Figure 5—figure supplement 4; see Materials and Methods for details). This indicates that human adult dermal fibroblasts have little to no ?-cell conversion potential. Despite repeated attempts, we were unable to assess the conversion property of human acinar cells due to their inability to grow in culture (Figure 1 CD133-negative population), consistent with other reports that primary acinar cells are difficult to culture. Thus, although we are unable to test all human cell types in this way, our work suggests that primary adult pancreatic ductal cells have a special latency for conversion toward an endocrine islet and beta cell-like fate.

To our knowledge, work by Sapir and colleagues (45) is the only report describing the use of human primary hepatocytes for β-cell transdifferentiation. In this report, however, the conversion toward an insulin-producing fate was comparatively poor; resulting cells produced about 10,000-fold lower insulin mRNA level than that of human islets, about 3–4 orders of magnitude lower than from conversion of pancreatic duct spheres. In addition, characteristic dense core vesicles in converted hepatocytes were not observed, indicating insufficient conversion towards ?-cells. In our report, we have presented evidence for acquisition of several characteristics of insulin-producing cells by converted duct spheres, including development of characteristic dense core vesicles, high Insulin content (estimated 49–77% total insulin in each insulin-producing cell compared to human islets), absence of glucagon or other islet hormone co-expression, modest glucose sensing and insulin secretion capacity, and other features detailed in our results.

These features also distinguish our work from studies reporting insulin-producing cells produced from in vitro differentiation of human embryonic stem cells (hESCs) and induced pluripotent stem cells (iPSCs). In our opinion, the qualitative and quantitative level of beta-cell phenotypes achieved by genetically-directed duct sphere conversion is higher than that achieved by prior hESC or iPSC studies in vitro. Moreover, our findings suggest that genetic methods may enhance beta-cell development from such cell lines

In sum, we believe conversion of human adult duct spheres is by far the most robust method yet reported for generating insulin-producing cells from human primary cells. We incorporated these points in the revised manuscript to clearly demonstrate the advantage of ductal cell use in human. We thank the reviewer for pointing this out.

*2) The final cocktail of transcription factors is stated to produce monohormonal insulin-producing cells, but this is not directly shown in the figures. This is an important point because many other differentiation assays generate fetal-like polyhormonal cells that cannot respond to glucose in the manner of adult beta cells. The cells produced here are not fully functional adult-type cells*.

We used immunochemistry to detect co-expression of C-peptide with GCG, PPY, and GHRL, and included such GCG and GHRL co-staining images in a new Figure 5. We found no PPY-positive cells from 7,487 GFP-positive cells we screened from two different samples. We found three GCG- positive cells out of 4,460 GFP-positive cells screened and none were co-labeled with C-peptide.

*3) How long does expression of the exogenous factors persist? Is it required for ongoing maintenance of the cells or can you demonstrate independence of the exogenous factors*?

Expression of the exogenous factors persisted after two weeks of maturation period (Figure 5—figure supplement 3) and at least 5 months, evidenced by the GFP-positive insulin-producing cells in transplanted mice (Figure 5—figure supplement 2). Transgenes delivered by adenovirus do not generally persist long term (67), especially in dividing cells. Persistent transgene expression noted in our study likely results from the known ability of Neurog3 to induce cell cycle exit (35), thereby preventing dilution of transgene-encoded factors by cell division. This precluded tests to demonstrate independence of the transgene-encoded factors for ongoing maintenance of the converted-cell phenotypes observed. We clarified this point in our revised manuscript.

*4) How sure are they that the starting population is pure ductal cells, given that CD133 is not exclusively expressed in ductal epithelium in the pancreas*?

In human pancreas, CD133 is expressed exclusively in ductal cells and centroacinar cells, but not other cell types in pancreas, including acinar or islet endocrine cells. This has been shown by us (Figure 1, Figure 1—figure supplement 1) and by others (Immervoll et al., BMC cancer, 2008; Lardon et al., Pancreas, 2008). In addition, we assessed the purity of the ductal cell fraction collected by qRT-PCR and immunostaining of CD133^+^ “sorted” cells (Figure 1). Thus, we are confident that CD133-based sorting can efficiently eliminate contaminating native islet endocrine cells, permitting conclusions about conversion toward endocrine cell fate. We understand that centroacinar cells may still be included in our CD133^+^ fraction. Therefore, we modified the phrase “purify ductal cells” to “fractionate ductal cells” in our revised manuscript.

Can they double sort with a general epithelial marker to further purify the population, given that CD133 only enriches 4-fold for sphere-forming cells?

Ductal cells already constitute large portion in unsorted cell population, ranging from 30% to 40% (8). Therefore, an average of 4-fold enrichment is expected even by achieving pure ductal cell isolations. Supporting this is our finding that the CD133-negative population is completely devoid of the sphere-forming cells (Figure 1).

Reviewer #2:

*In general this work is very well done, with convincing images and clear data*.

We thank the reviewer for this positive assessment.

*1) As CD133 is detected in centroacinar cells as well as in ductal cells (Immervoll et al JHC 2011), the authors should include additional acinar markers in their expression profile (*Figure 1*) to exclude acinar contamination*.

As requested, we performed qPCR with additional acinar marker carboxyl ester lipase (*CEL*) and included this data in Figure 1—figure supplement 1.

*2) Please include the average percentage of CD133*^*+*^
*cells detected in human pancreas and the purity of the sorted populations*.

As suggested, we calculated the average percentage of CD133^+^ fraction as 32.73% (n=32). We included this in our revised manuscript. To assess the purity of the sorted populations, we included qRT-PCR and immunostaining data of the “sorted” CD133+ and CD133-negative population with various markers in Figure 1.

*3)*
Figure 2
*please include co-staining of KRT19 and CD133*.

As suggested, we performed co-staining of KRT19 and CD133 on spheres and included these new data in Figure 2—figure supplement 1.

*4) The authors state “Time-lapse imaging revealed that spheres arose from single isolated CD133*^*+*^
*ductal cells”. However, this statement is not accurate unless the purity of the sort was 100%*.

We agree with this point. To eliminate any possible confusion, we modified the sentence to “Time-lapse imaging revealed that spheres arose from single cells.”

*5) Does the percentage of CD133*^*+*^
*cells decrease in culture? What is the percentage of CD133 after 3 months in culture*?

We used immunochemistry to quantify CD133 expression in cells comprising G1 and G7 spheres and quantified CD133^+^ cells. 95.0 ± 3.0% and 98.1 ± 0.58% cells were positive for CD133 in G1 and G7 spheres, respectively (n=6 sections each, ≥ 1300 cells counted). Rare CD133-negative cells are likely due to the exclusion of cell apical regions during tissue section, where CD133 expression is localized. We included this data in Figure 2—figure supplement 1 and E, and evaluated these results in our discussion.

*6) The authors state that Insulin*^*+*^
*IPCs did not express other islet hormones, the authors should include co-staining of c-peptide with GCG, PP and Ghrelin in*
Figure 5.

We agree and now include appropriate images of the requested co-immunostaining results. We performed co-staining of C-peptide with GCG, PPY, and GHRL, and included GCG and GHRL co-staining images in a new Figure 5. We found no PPY-positive cells from 7,487 GFP-positive cells we screened from two different samples. We found three GCG-positive cells out of 4,460 GFP-positive cells screened and none were co-labeled with C-peptide.

Reviewer #3:

*The claim that adult human pancreatic duct cells have a latent capacity for endocrine differentiation is correct, but only in this context of extremely strong transcription-factor-based enforced differentiation*.

We did not know if this comment required a response but for completeness, we included the following paragraph in our manuscript discussion.

“During mouse pancreas development, Neurog3 levels surges in a subset of pancreatic progenitors located in primitive ducts...”

*I always wonder what MNP6 would do to a non-pancreatic cell type, and therefore if this effect is a specific latency of pancreatic duct cells or not (the paragraph starting ‘The transcription factors MafA, Neurog3 and Pdx1…’ is more important as a result if there is something these cells can do that is not ever seen with the 4-factor combination MNP6 (4TF) or 4TFM)*.

Please see our response to the similar comment (#1) from the first reviewer.

In addition, in every experiment and analysis we performed, we included spheres (derived from CD133^+^ pancreatic ducts) infected with control viruses (Ad-RFP or Ad-GFP) but otherwise identical conditions as with N, 4TF or 4TFM spheres. We did not detect evidence of endocrine cell conversion by qRT-PCR, immunohistochemistry, and Insulin C-peptide ELISA (data from these controls are all included and labelled appropriate a “N”, “4TF” or “4TFM”). Therefore, we concluded that spontaneous conversion does not occur at detectable frequency.

*Essential is the claim that the insulin-producing cells are mono-hormone-positive, but this is not shown in the paper. The authors refer to the*
Figure 5
*as the one showing no double-positivities, but no Gcg, PP, or Ghrl are shown here. Some more clarity on this aspect seems critical. Assays for Gcg and other hormones are either referred to as data not shown, or this aspect is stated but the figure does not have the supporting data. Gcg immunodetection was done on cells from N alone? Gcg, PP were tested on the 4-factor combinations*?

We agree that the claim of mono-hormonal development is important to document. Please see also our response to a similar comment (#2) from the first reviewer. We were unable to detect mRNA or protein for *GCG* and *PPY* in Ad-RFP-Neurog3 infected spheres by qRT-PCR, now stated in the manuscript and in a revised Figure 4—figure supplement 1. For 4TFM spheres, we performed co- staining of C-peptide with GCG, PPY, and Ghrelin, and included GCG and GHRL co-staining images in a new Figure 5. We found no PPY-positive cells from 7,487 GFP-positive cells we screened from two different samples. We found three GCG-positive cells out of 4,460 GFP-positive cells screened and none were co-labeled with C-peptide. Thus we feel our claim of mono-hormonal insulin^+^ cells is well documented.

*Some explanation of why spontaneous (that is, non-TF-enforced) differentiation to endocrine fates was ruled out*.

We initially attempted to induce spontaneous differentiation of pancreatic ductal cells using systematic variations of culture conditions, but these efforts proved unsuccessful. This motivated us to induce Neurog3 to mimic the embryonic endocrine cell development explained above. We detailed this, as requested, in our revised manuscript.

In addition, as we described above, in every experiment and analysis we performed we included spheres (derived from pancreatic ducts) infected with control viruses (Ad-RFP or Ad-GFP) but otherwise identical conditions as with N, 4TF or 4TFM spheres. These controls served to monitor the possibility of spontaneous differentiation. Using the thorough analyses presented in our manuscript, we were unable to detect any sign of spontaneous endocrine cell differentiation in these controls. Therefore, we concluded that spontaneous conversion occurred at negligible frequency.

*Is Pax6 already expressed in the MNP-adnovirus cultures? If so, why is more needed*?

We thank the reviewer for this good question: Pax6 mRNA level in MNP spheres was less than 0.03% of levels detected in primary human islets. This finding promoted us to include this critical factor in our cocktail. To document this result, we have now included Pax6 qRT-PCR data in Figure 4—figure supplement 1 and in our revised manuscript.

*The part on “We sought new methods to mature the cells” (my words) reads strangely. It's the same method, just extending the time frame, I believe, and I would simplify this text*.

In addition to the extended time frame, we constructed new adenovirus permitting simultaneous expression of three factors (Pdx1, MafA, and Pax6) in a single virus, to increase the chance of these 3 factors being expressed in single cell. Therefore, we used only 2 viruses for 4TF (as opposed to 4 individual viruses for 4V). We clarified this in our revised manuscript.

*Another comment here is that we revert to the MNP6 mixture (4TF), having just been told that Pdx1 can be removed without impact – why*?

Even though exogenous PDX1 is not required for the conversion observed here, we found that exogenous PDX1 increased INS expression by two fold (Figure 4). This motivated us to include PDX1 in our MAFA-PAX6-PDX1 virus (Ad-GFP-M6P) construction.

Figure 2—figure supplement 1
*has an incorrect y-axis. 10,000 percent to the log(base10) is 4, correct. This graph needs altering – why not just “fold” for cell number? Related to this, Discussion asserts 3,000-fold, but this is just once? Up to 3,000-fold, and more typically xx-fold? Why do the cultures suddenly go bad at G7 (see text) – what happens – sudden apoptosis? #48 seems to be continuing even at G7 – please amend this text*.

We thank the reviewer for finding this error and we have amended the data presentation appropriately.

As was shown in Figure 2—figure supplement 1, we observed only one sample reached to G7. We agree with the reviewer and changed “>3,000-fold” to “up to 3,200-fold”.

We found the cells stopped growing and therefore could not be passaged in spheres after G7. We revised the text as suggested to “ductal cell expansion was not achieved, and spheres were not formed” in our manuscript.

*Does the 3-factor system (MNP) work in the authors' hands on mouse acini? And/or duct*?

In preliminary studies, we have found that the MNP combination can induce insulin expression in cultured mouse ductal cells. If deemed important by the editors and reviewers, we are happy to include this unpublished data.

*Does the CD133 separation method include centroacinar cells (CAC) or not? What is the photograph in*
Figure 1
*– ‘tip’ of duct dives out of plane of section before it is reached, and therefore the CAC cannot be seen in this panel? CAC could have a specific latency not seen in duct cells*.

We also think this is an interesting point and included this in our discussion. Briefly, CD133 is expressed in CACs in human pancreas, so we cannot exclude the possibility that CACs are included in CD133 sorted cell fraction. Indeed this is highly likely. We were unable to test the presence of CACs in our sorted CD133^+^ fraction due to a lack of specific “human” CAC marker. However, the sphere-forming efficiency approached 1 in 5 cells in our assay, and the frequency of CACs is thought to be much lower than this.

The idea that lineage-tracing methods are hard to apply in human cells should be stated, as everything else hinges on numerical arguments on cleanliness of cell separations, etc.

We have included a sentence in the Discussion to cover this point.

*chgA is an endocrine differentiation marker. This statement is meant to indicate that full-blown differentiation to the hormone-expressing state requires a substantial pulse of Ngn3. What about other hormones, even indicating non-pancreatic cell types*?

We performed qRT-PCR with CCK (Cholecystokinin) and GAST (Gastrin) – hormones expressed in the intestine – in Neurog3-overexpressed spheres and were unable to measure any detectable mRNA. We included this in our manuscript.

*A major point is the longevity of the pulse of Ngn3 and the other factors achieved with these methods, and the detection of a program that runs from the endogenous loci with or without the continuous presence of N, MNP, MNP6 (4TF, 4TFM). Would this method pulse the cells or not, and is continuous presence of some of the factors preventing their full differentiation to the most mature state*?

Expression of the exogenous factors persisted after two weeks of maturation period (Figure 5—figure supplement 3) and at least 5 months, evidenced by the GFP-positive insulin-producing cells in transplanted mice (Figure 5—figure supplement 2). Thus, the current method does not permit 'pulse' expression of the factors. Transgenes delivered by adenovirus do not generally persist long term (67), especially in dividing cells. Persistent transgene expression noted in our study likely results from the known ability of Neurog3 to induce cell cycle exit (35), thereby preventing dilution of transgenes and transgene-encoded factors by cell division. This precluded tests to demonstrate independence of the transgene-encoded factors for ongoing maintenance of the converted-cell phenotypes observed. We also agree with the reviewer that it is possible the continuous presence of some of the factors may prevent full differentiation toward more physiologically-mature endocrine cells. However, we do not feel these limitations change the main conclusions of our study. We have included a discussion of these points in our revised manuscript.

*Title of the section starting ‘Although ELISA studies readily detected Proinsulin production by IPCs…’ reads odd to me: ‘genetic conversion’ reads as if the genome of the adult duct cells is being altered in some manner*.

We agree with this reviewer’s point. We changed the phrase “genetic conversion” to “conversion”.

*Systematic removal of factors from MNP6 mixtures: Why can Pdx1 be removed without any impact*?

It is unclear why exogenous PDX1 is not critical for ductal sphere conversion toward an endocrine fate. One possibility is that our findings indicate distinct transcription factor requirements of human ductal epithelium (compared to mouse acinar cells and human hepatocytes that require Pdx1 expression for conversion, like in [67]). We included a discussion of this point in our revised manuscript.

*Various ‘obvious’ markers were not tested, or the manuscript is incorrect in not showing such ‘easily pointed out’ questions. Pdx1 is produced, by immunofluorescence assay, within these induced beta cells? To normal levels? MafA/B, etc? Nkx6.1 was assessed, but the ‘dogmatic’ mature β-cell TF list should be addressed, at least*.

We agree that MAFA and PDX1 are generally accepted adult β-cell markers in human. However, viral-expressed exogenous MAFA and PDX1 proteins from Ad-GFP-M6P adenovirus precluded us from detecting endogenous MAFA and PDX1 expression in IPCs. Based on this constraint, we used immunostaining to verify expression of other known ?-cell specific markers such as NKX6.1, IAPP, and PC1/3. We emphasized this point in our revised text.

*It does seem difficult to follow 4V, 4TF, 4TFM, MNP. Seems as if there is a mixed descriptor used for the same manipulation, at least sometimes; I suggest simply checking for a way of making the text fully consistent throughout*.

We have striven to achieve consistent use of these acronyms throughout the text in our revised manuscript.